# Studying the Effect of Introducing Chaotic Search on Improving the Performance of the Sine Cosine Algorithm to Solve Optimization Problems and Nonlinear System of Equations

Mohammed A. El-Shorbagy [1,2,*] and Fatma M. Al-Drees [1]

1 Department of Mathematics, College of Science and Humanities in Al-Kharj, Prince Sattam bin Abdulaziz University, Al-Kharj 11942, Saudi Arabia
2 Department of Basic Engineering Science, Faculty of Engineering, Menoufia University, Shebin El-Kom 32511, Egypt
* Correspondence: ma.hassan@psau.edu.sa

**Abstract:** The development of many engineering and scientific models depends on the solution of nonlinear systems of equations (NSEs), and the progress of these fields depends on their efficient resolution. Due to the disadvantages in solving them with classical methods, NSEs are amenable to modeling as an optimization issue. The purpose of this work is to propose the chaotic search sine cosine algorithm (CSSCA), a new optimization approach for solving NSEs. CSSCA will be set up so that it employs a chaotic search to get over the limitations of optimization techniques like a lack of diversity in solutions, exploitation's unfair advantage over exploration, and the gradual convergence of the optimal solution. A chaotic logistic map has been employed by many studies and has demonstrated its effectiveness in raising the quality of solutions and offering the greatest performance. So, it is used as a local search strategy. Three kinds of test functions—unimodal, multimodal, and composite test functions—as well as numerous NSEs—combustion problems, neurophysiology problems, arithmetic application, and nonlinear algebraic equations—were employed to assess CSSCA. To demonstrate the significance of the changes made in CSSCA, the results of the recommended algorithm are contrasted with those of the original SCA, where CSSCA's average improvement rate was roughly 12.71, demonstrating that it is very successful at resolving NSEs. Finally, outcomes demonstrated that adding a chaotic search to the SCA improves results by modifying the chaotic search's parameters, enabling better outcomes to be attained.

**Keywords:** optimization; chaotic search; sine cosine algorithm; nonlinear system of equations

**MSC:** 65K10; 68T20; 68U01; 68V20; 68W30; 90B99; 90C59

## 1. Introduction

The nonlinear system of equations serves as the foundation for many engineering and scientific models (NSEs), and finding a solution to this problem is crucial for the growth of these sectors. When practical models are converted into NSEs, NSEs can be discovered both directly and indirectly in some applications [1]. So, it is important to conduct a study into finding a robust and effective solution for NSEs.

Some of the methods that have traditionally been used to solve NSEs include the bisection technique, Muller's method, the false-position method, the steepest descent methods, the Broyden method, the Levenberg–Marquardt algorithm, the branch and prune approach, the Newton/damped Newton methods, Halley's method, and the Secant method [2]. The two best methods for resolving NSEs are Secant and Newton [3]. Other methods handle the NSEs by treating them as an optimization problem and applying the augmented Lagrangian method [4,5]. These methods take a long time, have a tendency to diverge, are ineffective when dealing with a group of nonlinear equations, and are

sensitive to the initial conditions. In order to create the Jacobian matrix, they also require a time-consuming procedure to calculate partial derivatives [6]. Finally, while dealing with problems that have a large size, they take a long time.

Meta-heuristic algorithms have been created as possible alternatives to numerical techniques to address their limitations as employing multiple initial guesses and depending on gradient information [7]. To enable strong exploration searchability and good exploitation, meta-heuristic algorithms were built on the foundation of randomization and local search. Additionally, they have additional features like simplicity and flexibility. There are four classifications of meta-heuristic optimization algorithms, which are illustrated in Figure 1 and include evolutionary, swarm, physical, behavioral based on humans, and biological algorithms [8–54].

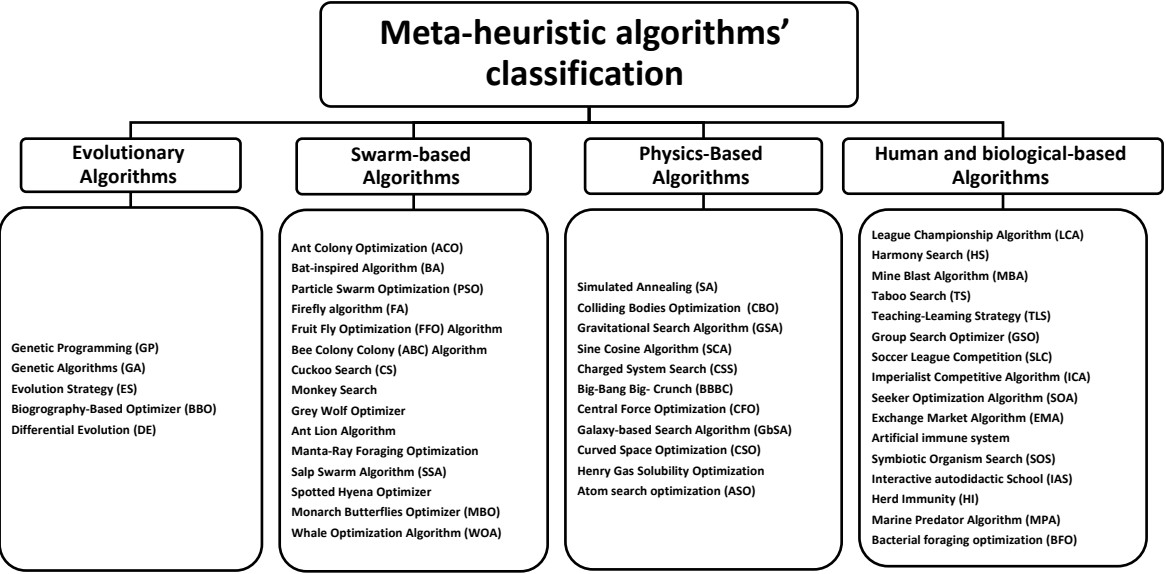

**Figure 1.** Meta-heuristic algorithms' classification.

Based on biological evolution, which includes reproduction, mutation, recombination, and selection, evolutionary-based algorithms are introduced. They adhere to the idea of survival based on suitability for a population of candidates (i.e., a collection of solutions) in a particular setting. Genetic programming (GP) [8], genetic algorithms (GA) [9], evolution strategy (ES) [10], biogeography-based optimizers (BBO) [11], and differential evolution (DE) [12] are some of the well-known techniques for evolutionary-based algorithms. The second classification is the swarm-based techniques that are inspired from flocking of birds. This behavior mimics how swarms interact with one another and with their surroundings. The popular swarm-based techniques are ant colony optimization (ACO) [13], bat-inspired algorithm (BA) [14], particle swarm optimization (PSO) [15], firefly algorithm (FA) [16], fruit fly optimization (FFO) algorithm [17], ant bee colony (ABC) algorithm [18], cuckoo search (CS) [19], monkey search [20], grey wolf optimizer (GWO) [21], ant lion algorithm (ALA) [22], manta-ray foraging optimization (MRFO) [23], salp swarm algorithm (SSA) [24], and others [25–27]. According to physical laws like electromagnetic force, gravitational force, inertia force, and others, the physical-based algorithms were offered to carry out communication and movement among the agents throughout the search space. Some of the algorithms in this classification are simulated annealing (SA) [28], colliding bodies optimization (CBO) [29], gravitational search algorithm (GSA) [30], sine cosine algorithm (SCA) [31], charged system search (CSS) [32], big-bang big-crunch (BBBC) [33], central force optimization (CFO) [34], and others [35–38]. Human-based algorithms are the fourth category that imitates human behavior as the league championship algorithm (LCA) [39], the harmony search (HS) algorithm [40], the mine blast algorithm (MBA) algorithm [41], the taboo search (TS) algorithm [42], the teaching-learning strategy (TLS) [43], the soccer league

competition (SLC) algorithm [45], the imperialist competitive algorithm (ICA) [46], the seeker optimization algorithm (SOA) [47], and the exchange market algorithm (EMA) [48]. The final classification of algorithms is one that is based on and inspired by biological behavior as the artificial immune system (AIS) [49] and bacterial forging optimization (BFO) [54].

The main characteristics of meta-heuristic algorithms are their exploration and exploitation searches, with exploration being in charge of finding the various interesting regions within the search space, while exploitation enhances the pursuit of ideal solutions inside a specified area. To achieve the best result during the search process, these features must be tuned. On the other hand, the stochastic nature of meta-heuristic algorithms makes it very challenging to balance these features. Furthermore, the effectiveness of a particular algorithm in solving a set of problems does not imply that it will be superior in handling another problems with various natures. This fact surely poses a difficult problem and can spur scientists and researchers to continue creating new meta-heuristic algorithms for resolving a wider range of practical issues [55].

Several meta-heuristic algorithms have been employed to resolve NSEs, including the GA [56], PSO [57], ABC [58], improved cuckoo CS algorithm [59], FA [60], and GOA [61]. Algelany and El-Shorbagy in [56] suggest a chaotic enhanced GA (CEGA). The CEGA is set up by using a new definition, which is chaotic noise, in order to address the inadequacies of optimization approaches, such as a lack of diversity in solutions, an imbalance between exploitation and exploration, and a sluggish convergence of the ideal solution. Mo and Liu [57] proposed a conjugate direction method (CDM) to PSO for addressing NSEs in order to enhance PSO and enable rapid optimization of high-dimensional optimization problems. The CDM and PSO are combined in this approach, helping PSO avoid local minima and efficiently optimize high-dimensional optimization problems. Jia and He [58] combined the ABC and PSO algorithms to propose the hybrid ABC technique for resolving NSEs. The hybrid algorithm combines the benefits of both strategies to solve the problem of entering a local or premature optimum. Additionally, Zhou and Li suggested an improved CSA to deal with the NSEs in [59]. They used a unique encoding technique that guarantees the supplied solution is feasible without changing the cuckoo's evolution. Also, Ariyaratne et al. offered augmented FA to tackle NSEs as an optimization issue in [60] with several advantages, such as: the ability to produce several root estimates at once and the removal of the requirement for initial assumptions, function continuity, or even differentiation. Finally, a modified GA based on the GOA is proposed in [61] to solve NSEs. In order to escape the local minimum and advance the search process, the improvements depended on specific mathematical presumptions and changing the scope of the GOA control parameter.

The SCA is a meta-heuristic algorithm developed by Mirjalili [31] for solving optimization issues. It uses a mathematical model based on trigonometric sine and cosine functions to find the optimal solution. Mirjalili [31] demonstrated that SCA outperforms other contemporary meta-heuristic algorithms in terms of efficiency. However, SCA, like any meta-heuristic algorithm, is dependent on adaptive and random factors. Therefore, a satisfactory solution is not always produced. Furthermore, there is no internal memory for SCA to keep track of previously discovered solutions. The shortcomings of SCA motivated the introduction of a new optimization method in this paper. Chaotic search sine cosine algorithm (CSSCA) is the name of the proposed optimization method. There are numerous optimization methods that used the chaotic mathematical strategy to get a better performance [62]. The chaotic mathematical strategy has drawn a lot of interest and has been used in a variety of fields, including optimization. The proposed CSSCA combines the chaotic search and SCA. This combination tries to improve SCA by addressing its shortcomings, including the lack of diversity in solutions, the disparity between exploitation and exploration, and the sluggish convergence of the optimal solution.

The main contributions of this study include the following:

(1) Introducing the chaotic search sine cosine algorithm (CSSCA), a novel method that combines chaotic search and SCA to resolve NSEs.

(2)  Utilizing chaotic search to enhance the SCA-obtained solution.

(3)  Numerous well-known functions and many NSEs are used to test CSSCA.

(4)  Demonstrate the outstanding performance of the proposed method with numerical results and prove it statistically.

(5)  Examining the introduction of the chaotic search on SCA and its effect on enhancing the outcomes by altering the chaotic search's parameters and attaining improved results.

The structure of the paper is as follows. Methods and Materials are covered in Section 2. Section 3 describes the proposed methodology in full. Section 4 shows the numerical results and discussions. Section 5 concludes with observations and recommendations with discussion.

## 2. Methods and Materials

An overview of nonlinear system of equations, SCA, and chaos theory is given in this section.

### 2.1. Nonlinear System of Equations (NSEs)

An NSE is described as follows in mathematics [60]:

$$\text{NSEs} = \begin{cases} f_1(z) = 0 \\ f_2(z) = 0 \\ \quad . \\ \quad . \\ f_Q(z) = 0 \end{cases} \tag{1}$$

where $z = (z_1, z_2, \ldots, z_n)$ is a vector with $n$ components that is a subset of $R^n$, and $f_q \forall q = 1, 2, \ldots, Q$ are the nonlinear functions that convert $z = (z_1, z_2, \ldots, z_n)$ from the $n$-dimensional space of $R^n$ to the real line. While some of the functions might not be linear, others might be. To solve NSEs, one must locate a solution where each of the aforementioned $Q$ functions equal zero [63].

**Definition 1.** *The solution $z = (z_1, z_2, \ldots, z_n)$ is referred to as the optimal solution of the NSEs if the functions $f_q(z) = 0$.*

By including the left side of all equations and utilizing the absolute value function as a constrained optimization problem, many methods [64,65] convert the NSEs into a problem that can be solved.

$$F(z) = abs(f_1(z) + f_2(z) + \cdots + f_Q(z)),$$
$$subject\ to = \begin{cases} f_1(z) = 0 \\ f_2(z) = 0 \\ \quad . \quad ; \\ \quad . \\ f_Q(z) = 0 \end{cases} \tag{2}$$

where $F(z)$ denotes the objective function. The objective function in (2) has a global minimum if all the nonlinear equations equal to 0.

### 2.2. Traditional SCA

The "No Free Lunch Theorems" (NLF theorem) [66] and the ongoing increase in real-world problems have inspired many academics to develop new algorithms. Depending on the problem type and its characteristics, various algorithms have been proposed in the literature. As a result, no universal method can solve every issue and ensure an overall solution. This truth is known as the NLF theorem. According to the NLF theorem, Algorithm A can outperform Algorithm B for some cost functions, while Algorithm B can outperform Algorithm A for other cost functions [67].

The mathematical formulas published by Mirjalili in 2016 served as the inspiration for SCA, a population-based algorithm [31]. The fundamental idea behind SCA is to use the behaviors of the sine and cosine functions in mathematics to do optimization searches. Like many optimization approaches, SCA begins with the initialization phase, in which a population of agents generates initial solutions in a random manner. Using some stochastic parameters and the properties of the sine and cosine functions, these agents are iteratively updated. The following is a description of the SCA working phases.

A.    **Updating Phase**

Each solution can be updated at this phase as follows:

$$x_i^{(t+1)} = \begin{cases} x_i^{(t)} + A_1 \cdot sin(A_2) \cdot \left| A_3 x_{best}^{(t)} - x_i^{(t)} \right| & r < 0.5 \\ x_i^{(t)} + A_1 \cdot cos(A_2) \cdot \left| A_3 x_{best}^{(t)} - x_i^{(t)} \right| & r \geq 0.5 \end{cases} ; \tag{3}$$

where $x_{best}^{(t)}$ denotes the best solution vector at the $t_{th}$ iteration and $x_i^{(t+1)}$, $x_i^{(t)}$ are the $i_{th}$ ($i = 1, 2, \ldots, n$) solution vectors at $(t+1)_{th}$ and $t_{th}$ iterations, respectively. Here, $r$ transitions from sine to cosine forms, and vice versa, using a uniformly distributed random number spread in the range (0, 1). Here, the movement of the present solution either moves towards the $x_{best}^{(t)}$ or outside the $x_{best}^{(t)}$ where it is controlled by the direction parameter $A_2$, while the $A_3$ weight parameter can emphasize the exploration approach ($A_3 > 1$) and increase the exploitation ($A_3 < 1$).

Figure 2 provides a conceptual explanation of the sine and cosine functions while updating the position within the interval $[-2, 2]$. The $A_2$ is inserted as you make this adjustment.

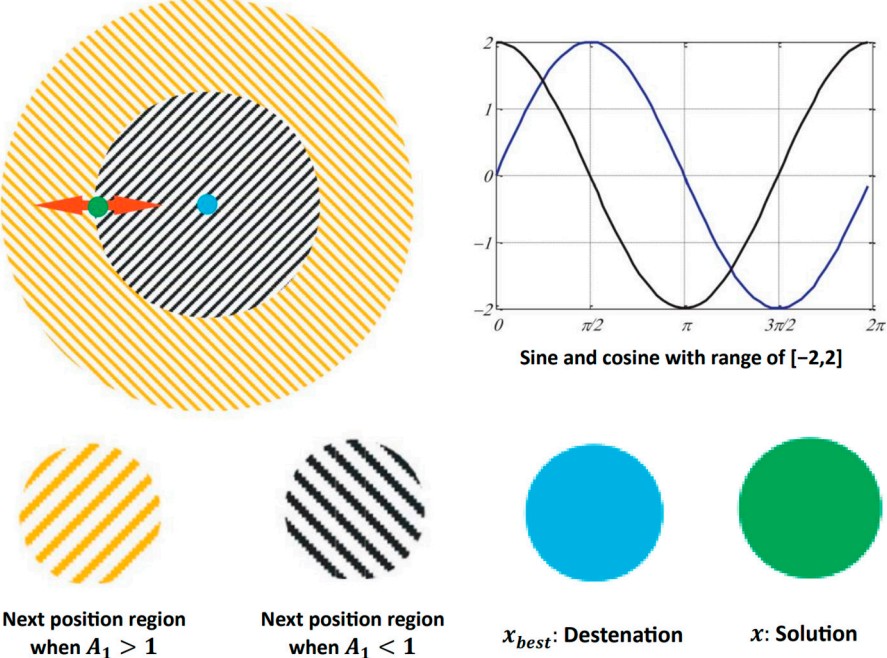

**Figure 2.** The procedure for updating in terms of sine and cosine functions with a [2, 2] range [31].

B.    **Balancing Phase**

The balancing phase is in charge of maintaining an acceptable balance between the exploration and exploitation in order to prevent the issue of premature convergence. In this method, the parameter $A_1$ can be introduced into the search space in the vicinity of the existing solution, which may be inside $x_{best}^{(t)}$ and $x_i^{(t)}$ or outside them. Hence, the parameter $A_1$ contributes to the exploration feature during the first half of the total number of iterations

and may encourage exploitation during the second half. The following mathematical formula can be used to express the parameter $A_1$.

$$A_1 = 2 - \frac{2 \cdot l}{L};$$ (4)

where the current and maximum iterations are defined, respectively, by $l$ and $L$.

*2.3. Chaos Theory*

Numerous areas of the optimization sciences have recently benefited from the application of chaos theory mathematics. Hénon [68] first discussed chaos theory, and Lorenz [69] simplified it. Chaos is a frequent nonlinear phenomenon in nature that correctly depicts the complexity of a system and can be used for optimization. Chaos fundamentally differs from statistical randomness in its capacity to efficiently search the search space of interest and enhance the effectiveness of optimization techniques.

As a novel method for global optimization, chaos-based optimization algorithms have attracted a lot of attention. A chaotic map is an evolution function that behaves chaotically in some way [70–72]. A discrete-time or continuous-time parameter can be used to parameterize maps. Iterated functions are the most common form for chaotic maps. The Appendix A contains several well-known chaotic maps from the literature, including the sinusoidal, singer, Chebyshev, sine, tent, Gauss, circle, piecewise, logistic, intermittency, Liebovitch, and iterative maps [73].

The logistic map improves the quality of the solutions and provides the best performance, according to the findings in [74]. This study uses it as a result.

**3. The Proposed Methodology**

The proposed approach, which combines the chaotic search (CS) method and the sine cosine algorithm (SCA), is presented in this subsection. The suggested approach comprises two phases. In the first phase, an approximative solution to the optimization issue is obtained using the optimization system SCA. The second phase then employs CS to speed convergence and improve the solutions' quality. The CSSCA fundamental steps can be summed up as follows:

**Phase 1: SCA**

**Step 1.** Agent $N$ is initially generated at random, and SCA parameters are maintained to create an initial population that meets the feasibility of the solved problem.

**Step 2.** The desired optimization fitness function is assessed for each agent.

**Step 3.** Set the best position $x_{best}^{(t)}$ and its objective value to the best initial agent's position and value for the first generation.

**Step 4.** According to Equation (3), search agents' positions are updated.

**Step 5.** Choose the population member with the best objective value as the best agent. Update $x_{best}^{(t)}$ and its objective value with the location and objective value of the current best agent if the objective value is superior to the objective value of $x_{best}^{(t)}$.

**Step 6.** The procedure is concluded, and the best result so far is returned as the SCA global optimum if the maximum number of generations have been produced, or when the population's agents converge. When every agent's position in the population is the same, convergence takes place. If not, go to Step 2 after updating the parameters of SCA ($A_1, A_2, A_3,$ and $r$).

Finally, SCA is used to generate an approximate solution, $x^* = (x_1^*, x_2^*, \ldots, x_n^*)$.

**Phase 2: Chaotic Search (CS)**

In the local region of $x^*$, which will be investigated, a chaos-based local search has the power to disrupt $x^*$. The following is a description of CS in more detail:

**Step 1.** Determine the variance range $[a_i, b_i]$, $i = 1, 2, \ldots, n$ of CS boundary by $x_i^* - \varepsilon < a_i$, $x_i^* + \varepsilon > b_i$.

**Step 2.** Create a chaos random number, $z_k$, by the logistic map as:

$$z_k = 4z_{k-1}(1 - z_{k-1}), \ z_0 \in (0,1), \ z_0 \notin \{0.0, 0.25, 0.50, \ 0.75, 1.0\} \tag{5}$$

**Step 3.** The variance range of the optimization valuable $[a_i, \ b_i]$ is mapped from the chaos variable $z_k$ as:

$$x_{i,k} = a_i + (b_i - a_i)z_k \tag{6}$$

which leads to:

$$x_{i,k} = x_i^* - \varepsilon + 2\varepsilon z_k \ \forall i = 1, \ldots, n. \tag{7}$$

**Step 4.** Set $x^* = x_k$ if $f(x_k) < f(x^*)$, otherwise breaking the iteration $k$.

**Step 5.** If $f(x^*)$ is not improved after all $k$ iterations, CS should be stopped and $x^*$ should be shown as the best option.

Figure 3 displays the proposed algorithm's flow chart.

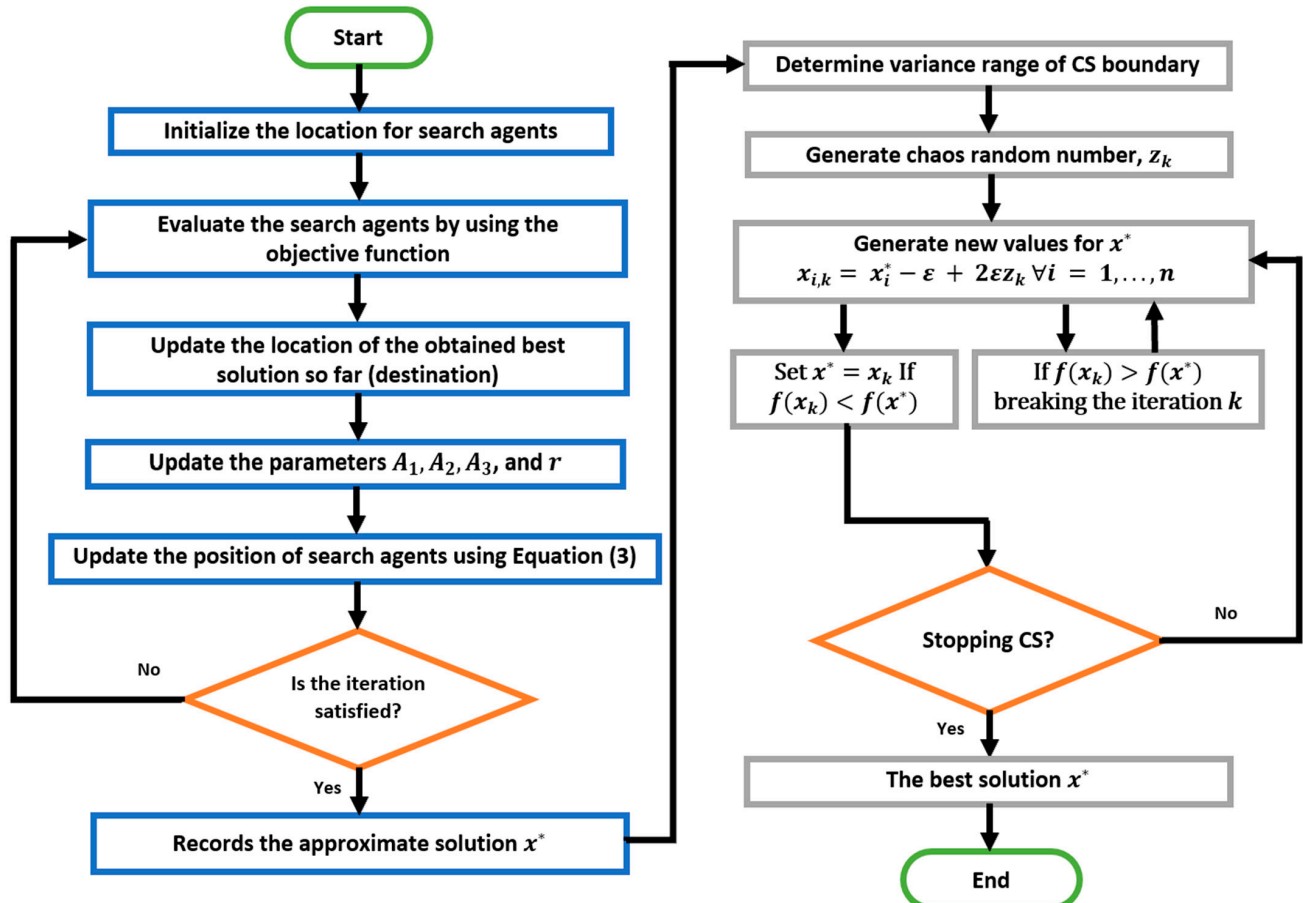

**Figure 3.** CSSCA's flowchart.

## 4. Numerical Results

This section assesses the performance of the suggested algorithm using 19 test functions. The mathematical formulation of these test functions is provided in Appendix B [31]. To show the advantages of the recommended changes, the outcomes are contrasted with those achieved by the original SCA and another hybrid algorithm called HGWOSCA that combines the GWO and SCA. To ensure that comparisons of algorithms are genuine, a statistical analysis of the results is carried out using the non-parametric Friedman test and the Wilcoxon signed-rank test. Seven SNEs are also resolved as a case study for CSSCA [75–77].

The suggested technique is written in MATLAB R2012b and put into practice on a computer running Windows 10 with an Intel(R) Core (TM) i7-6600U CPU running at 2.60 GHz and 16 GB of RAM.

Additionally, the CSSCA termination standard is described as

$$\delta = \|F_t\| - \|F_{t-1}\| \leq \varepsilon = 1e - 20 \tag{8}$$

$F_t$ is the estimated objective function at the iteration $t$, whereas $F_{t-1}$ the estimated objective function at the iteration $t - 1$.

*4.1. Results for 19 Test Functions*

These test functions are solved by the original SCA, HGWOSCA, and the proposed CSSCA. For computational studies, the agents' size $N$ is 1000, specified neighborhood radius $\varepsilon$ is 0.00001, and chaotic search iteration $k$ is 100,000.

Table 1 presents the comparison of results achieved by original SCA, HGWOSCA, and the proposed CSSCA. Table 1 confirms that CSSCA produces better solutions than that obtained by the original SCA and HGWOSCA on average. This indicates that entering CS on SCA caused SCA to perform clearly better and achieve positive results.

**Table 1.** Comparing the best function value obtained by the original SCA, HGWOSCA, and the proposed CSSCA.

| PD% between the Original SCA and CSSCA | CSSCA Result | HGWOSCA | SCA Result | PD% between the Original SCA and CSSCA |
|---|---|---|---|---|
| $F_1$ | 0.083239 | 0.0053 | 0.0535 | 35.72724324 |
| $F_2$ | $2.7645 \times 10^{-19}$ | 0.0319 | $2.7645 \times 10^{-19}$ | 0 |
| $F_3$ | $4.8556 \times 10^{-8}$ | $1.06612 \times 10^{-4}$ | $4.4013 \times 10^{-8}$ | 9.356207266 |
| $F_4$ | $1.3407 \times 10^{-10}$ | 0.7785 | $1.3407 \times 10^{-10}$ | 0 |
| $F_5$ | 7.3038 | 26.5837 | 7.0772 | 3.102494592 |
| $F_6$ | 0.80834 | 0.0031 | 0.6747 | 16.53264715 |
| $F_7$ | $4.6115 \times 10^{-4}$ | 0.0024 | $3.5722 \times 10^{-4}$ | 22.53713542 |
| $F_8$ | $-2049.352$ | $-5553.8$ | $-4272.3$ | 108.4707752 |
| $F_9$ | 6.1611 | 0 | 5.8554 | 4.961776306 |
| $F_{10}$ | $8.8818 \times 10^{-16}$ | 0.0026 | $8.8818 \times 10^{-16}$ | 0 |
| $F_{11}$ | 0.23479 | 0 | 0.2061 | 12.21943013 |
| $F_{12}$ | 0.084886 | 0.0025 | 0.0729 | 14.12011404 |
| $F_{13}$ | 0.21134 | 0.0011 | 0.2009 | 4.939907258 |
| $F_{14}$ | 1.0128 | 0.9980 | 0.9980 | 1.461295419 |
| $F_{15}$ | 0.00070061 | 0.0012 | $6.4760 \times 10^{-4}$ | 7.566263685 |
| $F_{16}$ | $-1.0316$ | $-1.0315$ | $-1.0316$ | 0 |
| $F_{17}$ | 0.40039 | 0.3979 | 0.3984 | 0.49701541 |
| $F_{18}$ | 3.0001 | 3 | 3.0000 | 0.003333222 |
| $F_{19}$ | $-3.8598$ | $-3.8625$ | $-3.8600$ | 0.005181616 |

In addition, to demonstrate the improvement between the original SCA and the new CSSCA algorithm, the following percentage decrease is used:

$$PD\% = \frac{|SCA\ Result - CSSCA\ Result|}{|SCA\ Result|} \cdot 100 \tag{9}$$

As indicated in Table 1's last column, CSSCA improved all results significantly by a 12.71% decrease on average. In other words, the aim of the hybrid between SCA and CS is to escape the local minimum and advance the search process, and the average improvement between CSSCA algorithm and the original SCA obtained by the PD% equation is 12.71% on average. So, this means that we can, therefore, conclude that CS directs SCA to remove the local minimum and improve the search results.

### 4.1.1. Friedman Test

The Friedman test is used to statistically rank the significance of algorithms [78]. Table 2 lists the test's outcomes in summary form. Since the *p*-value obtained in this statistical study is less than 0.05 $\alpha = 0.022$), there are significant differences between the CSSCA and the other two algorithms that are also examined. The CSSCA wins this statistical investigation where Figure 4's chart displays the rating of the CSSCA and competing algorithms. The smallest bar on the graph represents the best algorithm, while the longest bar depicts the worst. SCA earned the longest bar on the graph with a mean rank of 2.47, while CSSCA earned the shortest bar with a mean rank of 1.63. Hence, the graph shows that the CSSCA outperforms other algorithms by obtaining the top rank.

**Table 2.** Friedman test' results for the 19 test functions.

| Test Statistics | | Rank | |
|---|---|---|---|
| **N** | **19** | **Algorithm** | **Mean Rank** |
| Chi-Square | 7.657 | SCA | 2.47 |
| df | 2 | HGWOSCA | 1.89 |
| Asymp. Sig. | 0.022 | CSSCA | 1.63 |

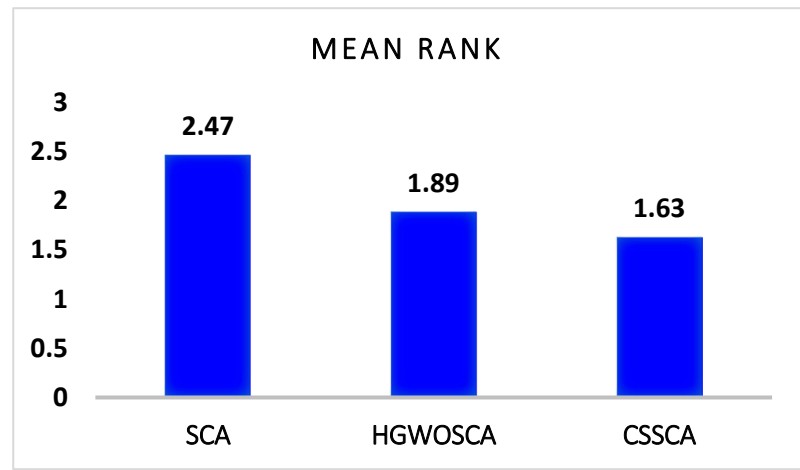

**Figure 4.** SCA, HGWOSCA, and CSSCA's mean rankings in the Friedman test.

### 4.1.2. Wilcoxon Signed-Rank Test

The CSSCA's distinction from the other algorithms is demonstrated using the Wilcoxon signed-rank test [78]. The Wilcoxon signed-rank test results are shown in Table 3. $R^+$ is the total of all positive ranks, whereas $R^-$ is the total of all negative ranks. By achieving $R^+$ values greater than $R^-$ values, CSSCA surpasses the other two algorithms in comparison, as demonstrated in Table 3. We can, therefore, conclude that CSSCA is a more effective algorithm than the other methods.

**Table 3.** The 19 test functions' Wilcoxon signed-ranks test outcomes.

| Test Statistics [a] a. Wilcoxon Signed-Ranks Test | | Ranks | | | | |
|---|---|---|---|---|---|---|
| | | Sum | N | Mean Rank | Sum of Ranks | < Or > Or = |
| SCA—CSSCA | | $R^-$ | 0 [a] | 0.00 | 0.00 | a. SCA < CSSCA |
| Z | $-3.408$ [b] | $R^+$ | 15 [b] | 8.00 | 120.00 | b. SCA > CSSCA |
| Asymp. Sig. (2-tailed) | 0.001 | Ties | 4 [c] | | | c. SCA = CSSCA |
| b. Based on negative ranks. | | Total | 19 | | | |
| HGWOSCA—CSSCA | | $R^-$ | 9 [d] | 10.67 | 96.00 | d. HGWOSCA < CSSCA |
| Z | $-0.923$ [c] | $R^+$ | 8 [e] | 7.13 | 57.00 | e. HGWOSCA > CSSCA |
| Asymp. Sig. (2-tailed) | 0.356 | Ties | 2 [f] | | | f. HGWOSCA = CSSCA |
| c. Based on negative ranks. | | Total | 19 | | | |

*4.2. Case Study: Solving NSEs*

The CSSCA algorithm will be used to solve several kinds of nonlinear systems of equations in this subsection. All the information of these nonlinear systems can be obtained in [75–77]. The NSEs are first transformed using Equation (2) into an optimization problem. The suggested method CSSCA then resolves this optimization problem. The descriptions of these NSEs are as follows:

1. **Case 1:** It contains the following two nonlinear algebraic equations.

$$F_1 = \begin{cases} f_1(x_1, x_2) = x_1 + x_2 + x_1^2 + x_2^2 - 8 = 0 \\ f_2(x_1, x_2) = x_1 + x_2 + x_1 x_2 - 5 = 0 \\ x_1 \in [-3.5,\ 2.5]\ , x_2 \in [-3.5,\ 2.5]\ . \end{cases} \tag{10}$$

2. **Case 2:** It contains the following three nonlinear equations:

$$F_2 = \begin{cases} f_1(x_1, x_2, x_3) = 3x_1^2 + \sin(x_1 x_2) - x_3^2 + 2 = 0 \\ f_2(x_1, x_2, x_3) = 2x_1^3 - x_2^2 - x_3 + 3 = 0 \\ f_3(x_1, x_2, x_3) = \sin(2x_1) + x_2 + \cos(x_2 x_3) - 1 = 0 \\ x_1 \in [-5,\ 5]\ ,\ x_2 \in [-1,\ 3]\ ,\ x_3 \in [-5,\ 5]\ . \end{cases} \tag{11}$$

3. **Case 3:** It contains non-differentiable system of non-linear equations as follows:

$$F_3 = \begin{cases} f_1(x_1, x_2) = x_1^2 - x_2 + 1 + \frac{1}{9}|x_1 - 1| = 0 \\ f_2(x_1, x_2) = x_2^2 - x_1 - 7 + \frac{1}{9}|x_2| = 0 \\ x_1 \in [-2,\ 2]\ , x_2 \in [1,\ 6]\ . \end{cases} \tag{12}$$

4. **Case 4:** It contains the following two nonlinear equations:

$$F_4 = \begin{cases} f_1(x_1, x_2) = \cos(2x_1) - \cos(2x_2) - 0.4 = 0 \\ f_2(x_1, x_2) = \sin(2x_2) - \sin(2x_1) + 2(x_2 - x_1) - 1.2 = 0 \\ x_1 \in [-10,\ 10]\ , x_2 \in [-10,\ 10]\ . \end{cases} \tag{13}$$

5.  **Case 5:** It is a combustion problem with a complex set of nonlinear equations, as shown below:

$$F_5 = \begin{cases} f_1 = x_2 + 2x_6 + x_9 + 2x_{10} - 10^{-5} = 0 \\ f_2 = x_3 + x_8 - 3 \cdot 10^{-5} = 0 \\ f_3 = x_1 + x_3 + 2x_5 + 2x_8 + x_9 + x_{10} - 5 \cdot 10^{-5} = 0 \\ f_4 = x_4 + 2x_7 - 10^{-5} = 0 \\ f_5 = 0.5140473 \cdot 10^7 x_5 - x_1^2 = 0 \\ f_6 = 0.1006932 \cdot 10^{-6} x_6 - 2x_2^2 = 0 \\ f_7 = 0.7816278 \cdot 10^{-15} x_7 - x_4^2 = 0 \\ f_8 = 0.1496236 \cdot 10^{-6} x_8 - x_1 x_3 = 0 \\ f_9 = 0.6194411 \cdot 10^{-7} x_9 - x_1 x_2 = 0 \\ f_{10} = 0.2089296 \cdot 10^{-14} x_{10} - x_1 x_2^2 = 0 \\ -10 \le x_1, x_2, \ldots, x_{10} \le 10. \end{cases} \tag{14}$$

6.  **Case 6:** It is a neurophysiology problem with a complex set of nonlinear equations, as illustrated below:

$$F_6 = \begin{cases} f_1 = x_1^2 + x_3^2 - 1 = 0 \\ f_2 = x_2^2 + x_4^2 - 1 = 0 \\ f_3 = x_5 x_3^3 + x_6 x_4^3 = 0 \\ f_4 = x_5 x_1^3 + x_6 x_2^3 = 0 \\ f_5 = x_5 x_1 x_3^2 + x_6 x_4^2 x_2 = 0 \\ f_6 = x_5 x_1^2 x_3 + x_6 x_2^2 x_4 = 0 \\ -10 \le x_1, x_2, \ldots, x_6 \le 10. \end{cases} \tag{15}$$

7.  **Case 7:** It is an arithmetic application that has a complex set of nonlinear equations, as illustrated below:

$$F_7 = \begin{cases} f_1 = x_1 - 0.254287220 - 0.18324757 \cdot x_4 x_3 x_9 = 0 \\ f_2 = x_2 - 0.378421970 - 0.16275449 \cdot x_1 x_{10} x_6 = 0 \\ f_3 = x_3 - 0.271625770 - 0.16955071 \cdot x_1 x_2 x_{10} = 0 \\ f_4 = x_4 - 0.198079140 - 0.15585316 \cdot x_7 x_1 x_6 = 0 \\ f_5 = x_5 - 0.441667280 - 0.19950920 \cdot x_7 x_6 x_3 = 0 \\ f_6 = x_6 - 0.146541130 - 0.18922793 \cdot x_8 x_5 x_{10} = 0 \\ f_7 = x_7 - 0.429371610 - 0.21180486 \cdot x_2 x_5 x_8 = 0 \\ f_8 = x_8 - 0.070564380 - 0.17081208 \cdot x_1 x_7 x_6 = 0 \\ f_9 = x_9 - 0.345049060 - 0.19612740 \cdot x_{10} x_6 x_8 = 0 \\ f_{10} = x_{10} - 0.426511020 - 0.21466544 \cdot x_4 x_8 x_1 = 0 \\ -10 \le x_1, x_2, \ldots, x_{10} \le 10. \end{cases} \tag{16}$$

Results for Nonlinear Systems of Equations

These NSEs cases are solved under various conditions for the chaotic search parameters to evaluate the effect of changing these parameters on the solution and its quality. These conditions are different values of the initial chaos random number $z_0$, the specified radius of chaotic search $\varepsilon$, and the iteration of chaotic search $k$ as follows:

$$[z_0, \varepsilon, k] = \begin{bmatrix} Conditions\ 1 \\ Conditions\ 2 \\ Conditions\ 3 \\ Conditions\ 4 \\ Conditions\ 5 \\ Conditions\ 6 \\ Conditions\ 7 \end{bmatrix} = \begin{bmatrix} (0.01, 0.01, 1000) \\ (0.01, 0.001, 1000) \\ (0.00001, 0.001, 1000) \\ (0.00001, 0.01, 1000) \\ (0.01, 0.01, 500) \\ (0.01, 0.01, 1500) \\ (0.01, 0.001, 1500) \end{bmatrix}. \tag{17}$$

The proposed CSSCA algorithm is a random approach as any meta-heuristic algorithm, where the improvement or accuracy is not guaranteed when solving any optimization

problem. So, these parameters were chosen randomly for the proposed algorithm to study the effect of changing it on the results. Tables 4–10 show the results of NSEs cases $(F_1 - F_7)$ at the different conditions, while Figures 5–11 show the graphical presentation of these Tables. From the tables and figures, we can see that the changing of the chaotic search parameters improves the quality of the solutions. In Cases 1 and 4, the best values of $F_1$ and $F_4$ occur at Condition 2 $(z_0, \varepsilon, k) = (0.01, 0.001, 1000)$, while in Case 2, the best value of $F_2$ occurs at condition 1 $(z_0, \varepsilon, k) = (0.01, 0.01, 1000)$. In Case 3, the best value of $F_3$ occurs at Condition 7 $(z_0, \varepsilon, k) = (0.01, 0.001, 1500)$, while in Cases 5 and 6, the best value of $F_5$ and $F_6$ occur at Condition 3 $(z_0, \varepsilon, k) = (0.00001, 0.001, 1000)$. Finally, in Case 7, the best value of $F_7$ occurs at all conditions except Condition 6 $(z_0, \varepsilon, k) = (0.01, 0.01, 1500)$. Finally, Figure 12 shows the convergence of $F$ for all NSEs cases with the chaotic search parameters. We notice from the figure that the introduction of the chaotic search on the SCA improves the results by changing the parameters of the chaotic search and thus better results can be obtained.

**Table 4.** Results of case 1: $F_1$ at different conditions.

| Conditions | $z_0$ | $\varepsilon$ | $k$ | Best Position $(x_1, x_2)$ | Best $F_1$ |
|---|---|---|---|---|---|
| Conditions 1 | 0.01 | 0.01 | 1000 | 2.0002, 0.9999 | $2.7336 \times 10^{-4}$ |
| Conditions 2 | 0.01 | 0.001 | 1000 | 1.0000, 2.0000 | $2.8633 \times 10^{-5}$ |
| Conditions 3 | 0.00001 | 0.001 | 1000 | 1.9996, 1.0007 | $6.6914 \times 10^{-4}$ |
| Conditions 4 | 0.00001 | 0.01 | 1000 | 2.0004, 0.9995 | $5.4087 \times 10^{-4}$ |
| Conditions 5 | 0.01 | 0.01 | 500 | 1.0006, 1.9996 | $6.0820 \times 10^{-4}$ |
| Conditions 6 | 0.01 | 0.01 | 1500 | 1.0001, 2.0000 | $4.0604 \times 10^{-4}$ |
| Conditions 7 | 0.01 | 0.001 | 1500 | 2.0006, 0.9991 | $8.6317 \times 10^{-4}$ |

**Table 5.** Results of Case 2: $F_2$ at different conditions.

| Conditions | $z_0$ | $\varepsilon$ | $k$ | Best Position $(x_1, x_2, x_3)$ | Best $F_2$ |
|---|---|---|---|---|---|
| Conditions 1 | 0.01 | 0.01 | 1000 | −0.0329, 1.2648, 1.4006 | $2.6693 \times 10^{-4}$ |
| Conditions 2 | 0.01 | 0.001 | 1000 | −0.0230, 1.2645, 1.4009 | $9.0229 \times 10^{-4}$ |
| Conditions 3 | 0.00001 | 0.001 | 1000 | −0.0333, 1.2627, 1.4056 | $8.3069 \times 10^{-4}$ |
| Conditions 4 | 0.00001 | 0.01 | 1000 | −0.0339, 1.2649, 1.3999 | $7.7486 \times 10^{-4}$ |
| Conditions 5 | 0.01 | 0.01 | 500 | −0.0338, 1.2650, 1.4002 | $7.3606 \times 10^{-4}$ |
| Conditions 6 | 0.01 | 0.01 | 1500 | −0.0338, 1.2648, 1.4001 | $6.1370 \times 10^{-4}$ |
| Conditions 7 | 0.01 | 0.001 | 1500 | −0.0336, 1.2650, 1.3998 | $7.4714 \times 10^{-4}$ |

**Table 6.** Results of Case 3: $F_3$ at different conditions.

| Conditions | $z_0$ | $\varepsilon$ | $k$ | Best Position $(x_1, x_2)$ | Best $F_3$ |
|---|---|---|---|---|---|
| Conditions 1 | 0.01 | 0.01 | 1000 | −1.3659, 3.1273 | $6.4579 \times 10^{-4}$ |
| Conditions 2 | 0.01 | 0.001 | 1000 | −1.3657, 3.1273 | $4.3137 \times 10^{-4}$ |
| Conditions 3 | 0.00001 | 0.001 | 1000 | 1.4419, 3.1273 | $5.3103 \times 10^{-4}$ |
| Conditions 4 | 0.00001 | 0.01 | 1000 | −1.3657, 3.1273 | $4.0623 \times 10^{-4}$ |
| Conditions 5 | 0.01 | 0.01 | 500 | 1.4417, 3.1273 | $1.8692 \times 10^{-4}$ |
| Conditions 6 | 0.01 | 0.01 | 1500 | 1.4419, 3.1273 | $4.0073 \times 10^{-4}$ |
| Conditions 7 | 0.01 | 0.001 | 1500 | 1.4416, 3.1273 | $1.4685 \times 10^{-5}$ |

**Table 7.** Results of Case 4: $F_4$ at different conditions.

| Conditions | $z_0$ | $\varepsilon$ | $k$ | Best Position $(x_1, x_2)$ | Best $F_4$ |
|---|---|---|---|---|---|
| Conditions 1 | 0.01 | 0.01 | 1000 | 0.1570, 0.4935 | $7.8956 \times 10^{-4}$ |
| Conditions 2 | 0.01 | 0.001 | 1000 | 0.1565, 0.4934 | $4.7106 \times 10^{-6}$ |
| Conditions 3 | 0.00001 | 0.001 | 1000 | 0.1566, 0.4934 | $7.0050 \times 10^{-5}$ |
| Conditions 4 | 0.00001 | 0.01 | 1000 | $-2.9851, -2.6482$ | $6.6633 \times 10^{-5}$ |
| Conditions 5 | 0.01 | 0.01 | 500 | 6.4402, 6.7768 | $8.2988 \times 10^{-4}$ |
| Conditions 6 | 0.01 | 0.01 | 1500 | 0.1568, 0.4935 | $4.0707 \times 10^{-4}$ |
| Conditions 7 | 0.01 | 0.001 | 1500 | 0.1570, 0.4936 | $7.1143 \times 10^{-5}$ |

**Table 8.** Results of Case 5: $F_5$ at different conditions.

| Conditions | $z_0$ | $\varepsilon$ | $k$ | Best Pos $(x_1, \ldots, x_{10})$ | Best $F_5$ |
|---|---|---|---|---|---|
| Conditions 1 | 0.01 | 0.01 | 1000 | $-5.0135 \times 10^{-6}$, $2.8680 \times 10^{-5}$, $-10$, 0.0037, $6.0578 \times 10^{-12}$, 10, $-0.0018$, 10, $-3.4240 \times 10^{-6}$, $-10$ | $3.9154 \times 10^{-4}$ |
| Conditions 2 | 0.01 | 0.001 | 1000 | 0.0038, $-0.0054$, $-0.0104$, $-0.0044$, $2.1600 \times 10^{-11}$, 0.0095, 0.0019, 0.0145, $-0.0307$, 0.0082 | $2.7376 \times 10^{-4}$ |
| Conditions 3 | 0.00001 | 0.001 | 1000 | $-1.1033 \times 10^{-8}$, $-3.1127 \times 10^{-6}$, $-10$, $-5.5679 \times 10^{-4}$, $-1.8332 \times 10^{-12}$, 10, $2.8015 \times 10^{-4}$, 10, $2.1671 \times 10^{-5}$, $-10$ | $6.0646 \times 10^{-5}$ |
| Conditions 4 | 0.00001 | 0.01 | 1000 | $-0.0104$, $-1.7648 \times 10^{-4}$, $-0.0222$, $3.1607 \times 10^{-4}$, $1.2687 \times 10^{-10}$, $-0.0152$, $-5.5569 \times 10^{-5}$, 0.0251, $-0.0671$, 0.0487 | $2.6557 \times 10^{-4}$ |
| Conditions 5 | 0.01 | 0.01 | 500 | $1.3712 \times 10^{-7}$, $-2.1896 \times 10^{-4}$, $-9.9991$, $-7.0970 \times 10^{-7}$, $-1.1578 \times 10^{-13}$, 10, $-1.4509 \times 10^{-5}$, 10, $-2.5461 \times 10^{-4}$, $-10$ | $3.4767 \times 10^{-4}$ |
| Conditions 6 | 0.01 | 0.01 | 1500 | $1.5492 \times 10^{-7}$, $-2.7488 \times 10^{-4}$, $-10$, $-4.9143 \times 10^{-6}$, $-2.9417 \times 10^{-12}$, 10, $3.0756 \times 10^{-6}$, 10, $2.9057 \times 10^{-4}$, $-10$ | $3.5273 \times 10^{-4}$ |
| Conditions 7 | 0.01 | 0.001 | 1500 | $1.2119 \times 10^{-6}$, $-3.0283 \times 10^{-4}$, 10, $3.6210 \times 10^{-5}$, $2.3696 \times 10^{-12}$, $-10$, $-1.1715 \times 10^{-5}$, $-10$, $3.0054 \times 10^{-4}$, 10 | $3.5469 \times 10^{-4}$ |



**Table 9.** Results of Case 6: $F_6$ at different conditions.

| Conditions | $z_0$ | $\varepsilon$ | $K$ | Best Pos ($x_1, \ldots, x_6$) | Best $F_6$ |
|---|---|---|---|---|---|
| Conditions 1 | 0.01 | 0.01 | 1000 | 0.1933, −0.9973, −0.9804, 0.0730, −5.0824 × 10$^{-4}$, 0.0031 | 8.7958 × 10$^{-4}$ |
| Conditions 2 | 0.01 | 0.001 | 1000 | 0.1431, 0.9524, −0.9896, 0.3050, 7.5318 × 10$^{-4}$, −0.0023 | 6.0153 × 10$^{-4}$ |
| Conditions 3 | 0.00001 | 0.001 | 1000 | −1.0000, 0.9687, 0.0030, −0.2495, −0.0017, −0.0013 | 2.4897 × 10$^{-4}$ |
| Conditions 4 | 0.00001 | 0.01 | 1000 | 1.7130 × 10$^{-5}$, −0.9636, 1.0019, 0.2684, 3.3332 × 10$^{-4}$, 4.3633 × 10$^{-5}$ | 7.9744 × 10$^{-4}$ |
| Conditions 5 | 0.01 | 0.01 | 500 | 0.7161, 1.0010, −0.6963, −0.0190, −1.5278 × 10$^{-4}$, 4.4880 × 10$^{-6}$ | 8.2790 × 10$^{-4}$ |
| Conditions 6 | 0.01 | 0.01 | 1500 | 0.1862, 0.6602, −0.9835, 0.7511, 0.0015, 8.7707 × 10$^{-4}$ | 6.8913 × 10$^{-4}$ |
| Conditions 7 | 0.01 | 0.001 | 1500 | 0.0092, 0.0114, 1.0000, 1.0013, −9.9631 × 10$^{-4}$, −0.0012 | 8.3455 × 10$^{-4}$ |

**Table 10.** Results of Case 7: $F_7$ at different conditions.

| Conditions | $z_0$ | $\varepsilon$ | $k$ | Best Pos ($x_1, \ldots, x_6$) | Best $F_7$ |
|---|---|---|---|---|---|
| Conditions 1 | 0.01 | 0.01 | 1000 | 0.2317, 0.3962, 0.2888, 0.2005, 0.4093, 0.1540, 0.4427, 0.0634, 0.2999, 0.4428 | 0.0176 |
| Conditions 2 | 0.01 | 0.001 | 1000 | 0.2317, 0.3962, 0.2888, 0.2005, 0.4093, 0.1540, 0.4427, 0.0634, 0.2999, 0.4428 | 0.0176 |
| Conditions 3 | 0.00001 | 0.001 | 1000 | 0.2317, 0.3962, 0.2888, 0.2005, 0.4093, 0.1540, 0.4427, 0.0634, 0.2999, 0.4428 | 0.0176 |
| Conditions 4 | 0.00001 | 0.01 | 1000 | 0.2317, 0.3962, 0.2888, 0.2005, 0.4093, 0.1540, 0.4427, 0.0634, 0.2999, 0.4428 | 0.0176 |
| Conditions 5 | 0.01 | 0.01 | 500 | 0.2317, 0.3962, 0.2888, 0.2005, 0.4093, 0.1540, 0.4427, 0.0634, 0.2999, 0.4428 | 0.0176 |
| Conditions 6 | 0.01 | 0.01 | 1500 | 0.2066, 0.4182, 0.2583, 0.1698, 0.4791, 0.1494, 0.4275, 0.0728, 0.3549, 0.4234 | 0.0190 |
| Conditions 7 | 0.01 | 0.001 | 1500 | 0.2317, 0.3962, 0.2888, 0.2005, 0.4093, 0.1540, 0.4427, 0.0634, 0.2999, 0.4428 | 0.0176 |

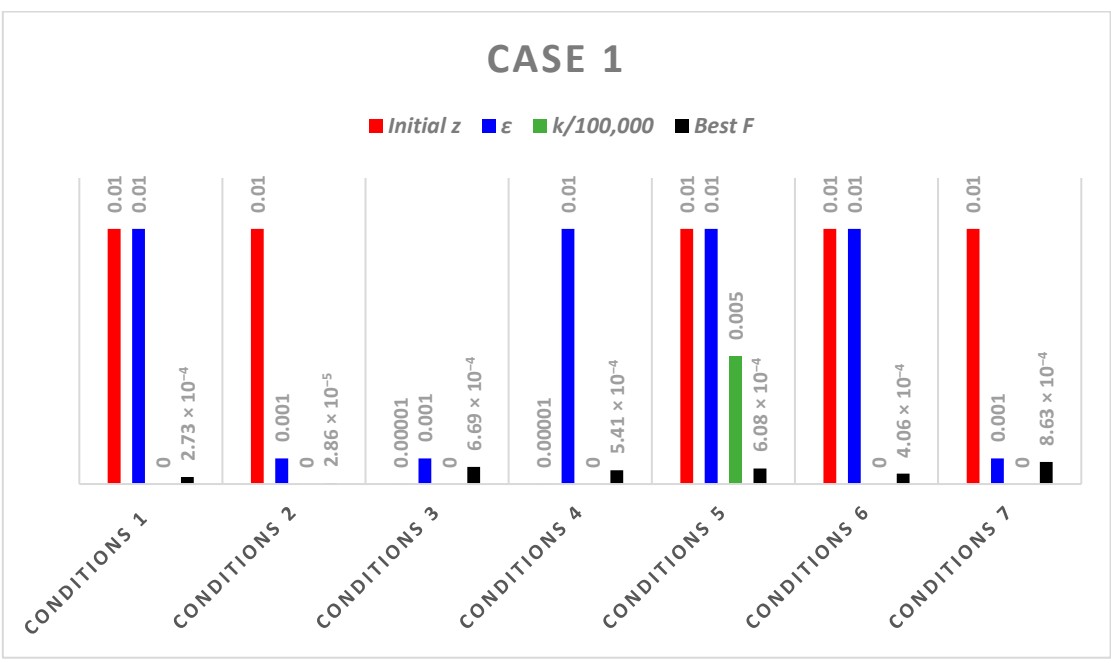

**Figure 5.** Graphical presentation of Table 4.

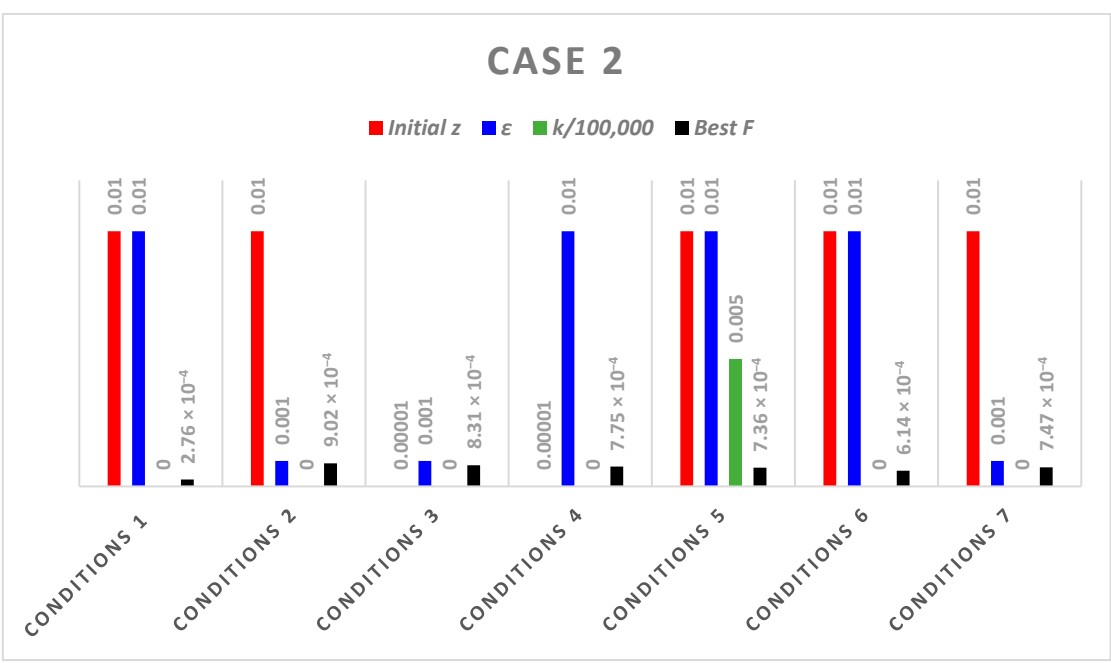

**Figure 6.** Graphical presentation of Table 5.

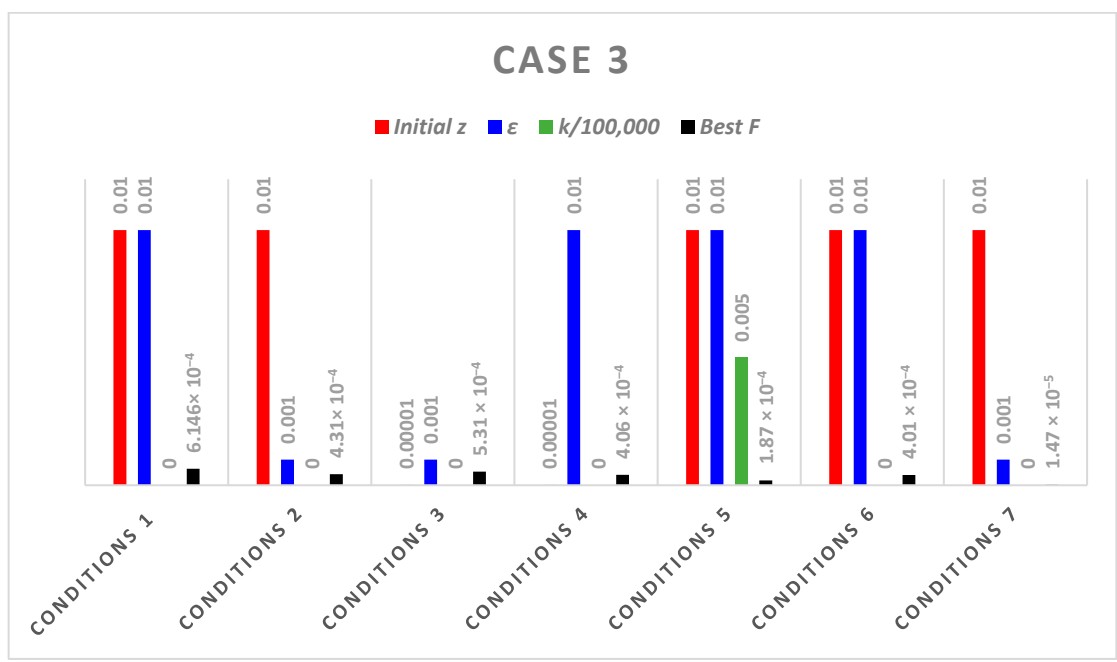

**Figure 7.** Graphical presentation of Table 6.

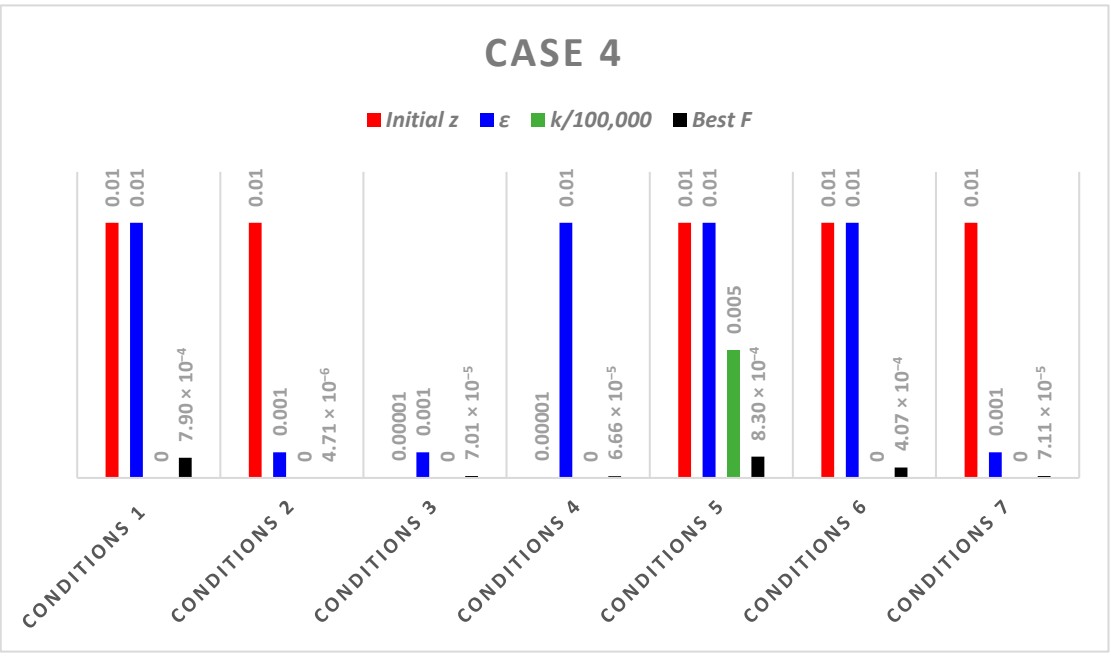

**Figure 8.** Graphical presentation of Table 7.

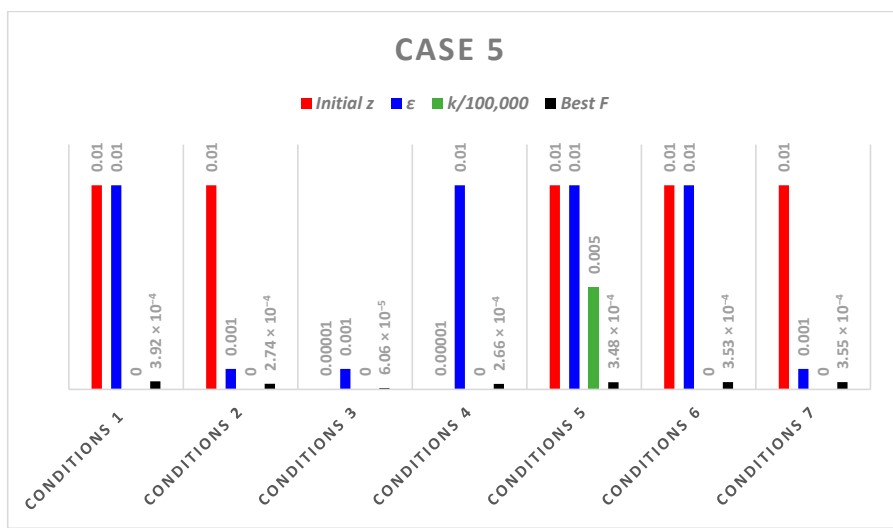

**Figure 9.** Graphical presentation of Table 8.

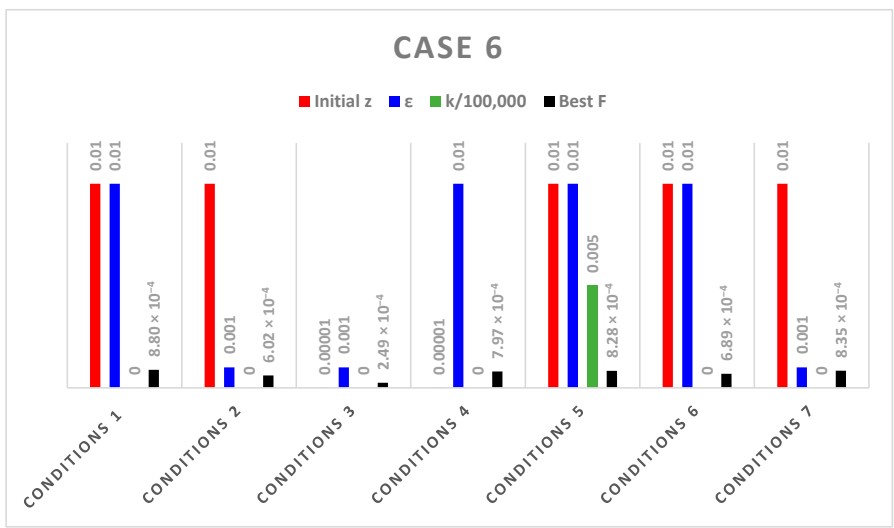

**Figure 10.** Graphical presentation of Table 9.

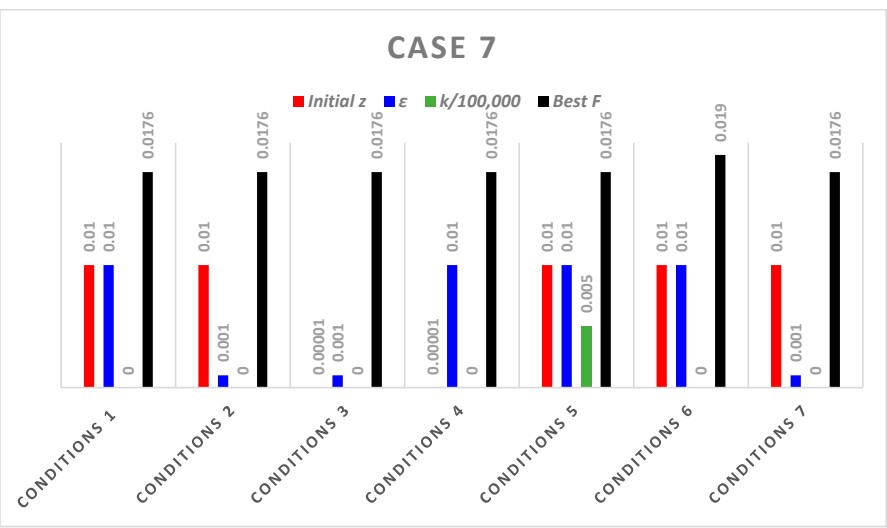

**Figure 11.** Graphical presentation of Table 10.

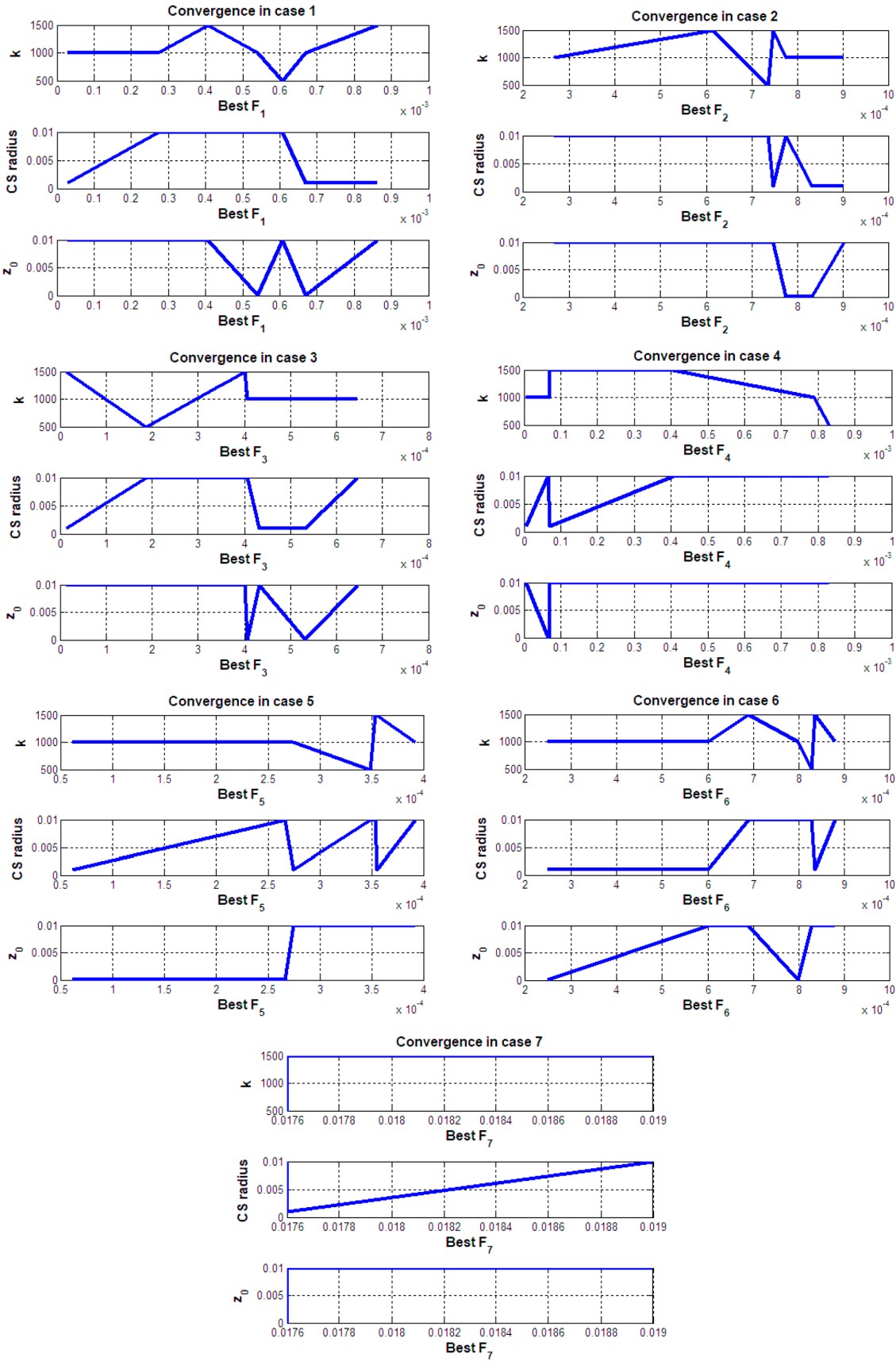

**Figure 12.** The convergence of *F* for all NSEs cases with the chaotic search parameters.

*4.3. Computational Complexity of the CSSCA*

When assessing any metaheuristic optimization technique's processing time, it is essential to take into account the computational complexity, which is influenced by the design and implementation of the algorithm. It should be emphasized that the initialization procedure, the assessment of the fitness function, and the updating of the solutions all play major roles in determining the computing cost of the proposed CSSCA. The initialization process has an $O(N, n)$ complexity, where $N$ represents the population size and $n$ represents the number of parameters in the problem (dimension). The complexity of updating solutions via SCA is $O(t_{max} \cdot N \cdot n)$, where $t_{max}$ represents the maximum number of SCA iterations. The updating solutions' complexity by CS is $O(k_{max} \cdot n)$; where $k_{max}$ indicates maximum iterations of CS. The computational complexity of updating solutions is thus $O(t_{max} \cdot N \cdot n) + O(k_{max} \cdot n)$. As a result, the suggested CSSCA has a computational complexity that is $O(N, n) + O(t_{max} \cdot N \cdot n) + O(k_{max} \cdot n)$. In the following section, several benchmark test functions are used to check and corroborate the proposed CSSCA's performance in addressing optimization difficulties.

## 5. Discussion and Conclusions

In this paper, a chaotic search sine cosine algorithm (CSSCA) is proposed to study the effect of introducing chaotic search (CS) on the original sine cosine (SCA) algorithm to improve its performance in terms of reaching a better solution. The advantages of both SCA and CS are combined in CSSCA, where the capabilities of SCA exploration and the CS exploitation are integrated. This combination also seeks to increase search effectiveness and avoid the local minimum by achieving the good balance between capabilities of exploration and exploitation.

The research proposed was divided into three parts. The proposed algorithm's performance was assessed using 19 test functions in the first part to demonstrate how the CS input influences its ability to be a better SCA performer. The second part aimed to show how to convert the nonlinear system of equations (NSEs) into an optimization problem. Then, this optimization problem of NSEs was solved with the CSSCA. In, the final Part, 7 NSE benchmarking problems were solved and studied how modifying the chaotic search parameters affected the solution's quality.

The following are the main study findings for the suggested algorithm:

(1) According to the results that were obtained in Table 1, we see those results of CSSCA are better than those obtained by the original SCA and the other SCA-based algorithm (HGWOSCA).

(2) The results obtained in Table 1, by the percentage decrease equation (Equation (9)), shows that adding CS to SCA improves the original SCA results by 12.71%. So, we can therefore infer that CS instructs SCA to get rid of the local minimum and optimize the search results for a better solution.

(3) Friedman and Wilcoxon's tests were used for the statistical analysis, and the findings are shown in Tables 2 and 3. Based on the results, it can be shown that the CSSCA and the other two algorithms that were also tested have significant differences, with a $p$-value of less than 0.05 ($\alpha = 0.022$). In addition, Figure 4 demonstrates that the CSSCA surpasses other algorithms by obtaining the first rank. Furthermore, Table 3 shows that CSSCA performs better than the other two algorithms since its $R^+$ values are larger than its than its $R^-$ values. This indicates that CSSCA performs better according to achieve lower objective function values for most testing functions.

(4) The results of NSEs in Tables 4–10 and Figures 5–11 show that the introduction of CS on the SCA affects its performance as it was found that changing the CS parameters has an impact on the quality of the solution and obtaining better results.

The suggested algorithm's possible flaw, like that of all meta-heuristic algorithms, is that there is no assurance that optimization problems will be solved without an increase to the time of computing or accuracy. This is because meta-heuristic algorithms employ random operations and chaotic search. To assess the effectiveness of the algorithm and

identify any potential weaknesses, other larger and more complex NSEs may be taken into consideration in further works. In addition, more experiments must be done to reach the optimal values for the parameters of the CS through which the optimal solution can be obtained when solving any optimization problem. Finally, it is recommended that optimization problems and NSEs be resolved using various optimization techniques, including the monarch butterfly optimization (MBO) [79], gradient-based optimizer (GBO) [80], beluga whale optimization (BWO) [81], wild horse optimizer (WHO) [82], etc.

**Author Contributions:** Conceptualization, M.A.E.-S. and F.M.A.-D.; Methodology, M.A.E.-S. and F.M.A.-D.; Writing and original draft preparation, F.M.A.-D.; Co-review and validation, M.A.E.-S.; Writing—editing; M.A.E.-S. All authors have read and agreed to the published version of the manuscript.

**Funding:**  The authors extend their appreciation to the Deputyship for Research & Innovation, Ministry of Education in Saudi Arabia for funding this research work through the project number (IF2/PSAU/2022/01/20208).

**Institutional Review Board Statement:** Not applicable.

**Informed Consent Statement:** Not applicable.

**Data Availability Statement:** All data used to support the findings of this study are included within the article.

**Conflicts of Interest:** The authors declare no conflict of interest.

**Appendix A. Chaotic Maps [73]**

– Sinusoidal map: The equation that produces the sine wave in Sinusoidal map is

$$x_{t+1} = ax_t^2 \sin(\pi x_t) \tag{A1}$$

where $a = 2.3$.

– Chebyshev map: The Chebyshev map is shown as

$$x_{t+1} = \cos\left(t\cos^{-1}(x_t)\right) \tag{A2}$$

– Singer map: The formulation of the one-dimensional chaotic Singer map is as follows:

$$x_{t+1} = \mu\left(7.86\, x_t - 23.31\, x_t^2 + 28.75\, x_t^3 - 13.302875\, x_t^4\right) \tag{A3}$$

where $\mu \in (0.9, 1.08)$.

– Tent map: The following iterative equation defines the tent map:

$$x_{t+1} = \begin{cases} \frac{x_t}{0.07}, & x_t < 0.7, \\ \frac{10}{3}(1 - x_t), & x_t \geq 0.7. \end{cases} \tag{A4}$$

– Sine map: As an example, consider the sine map:

$$x_{t+1} = \frac{a}{4}\sin(\pi x_t) \tag{A5}$$

where $0 < a \leq 4$.

– Circle map: According to the following typical equation, a circle map is:

$$x_{t+1} = x_t + b - (a - 2\pi)\sin(2\pi x_t) \tag{A6}$$

where $a = 0.5$ and $b = 0.2$.

–    Piecewise map: The formulation of the piecewise map is as follows:

$$x_{t+1} = \begin{cases} \frac{x_t}{p}, & 0 < x_t < p \\ \frac{x_t - p}{0.5 - p}, & p \leq x_t < 0.5 \\ \frac{(1 - p - x_t)}{0.5 - p}, & 0.5 \leq x_t < 1 - p \\ \frac{(1 - x_t)}{p}, & 1 - p < x_t < 1 \end{cases} \tag{A7}$$

where $p \in (0, 0.5)$ and $x \in (0, 1)$.

–    Gauss map: A nonlinear iterated map of the reals into a real interval determined by the Gaussian function is called the Gauss map, often referred to as the Gaussian map or mouse map:

$$x_{t+1} = exp\left(-\alpha x_t^2\right) + \beta; \tag{A8}$$

where $\alpha$ and $\beta$ are real parameters.

–    Logistic map: Without the need for any random sequence, the logistic map illustrates how complicated behavior can develop from a straightforward deterministic system. Its foundation is a straightforward polynomial equation that captures the dynamics of a biological population.

$$x_{t+1} = cx_t(1 - x_t); \tag{A9}$$

where $x_0 \in (0, 1), x_0 \notin \{0.0, 0.25, 0.50, 0.75, 1.0\}$ and when $c = 4.0$ The logistic map creates a chaotic sequence.

–    Intermittency map: Two iterative equations are used to create the intermittency map, which is shown as:

$$x_{t+1} = \begin{cases} \varepsilon + x_t + cx_t^n, & if\ 0 < x_t \leq p \\ \frac{x_t - p}{1 - p}, & else\ if\ p < x_t < 1 \end{cases} \tag{A10}$$

where $c = \frac{1 - \varepsilon - p}{p^2}, n = 2.0$ and $\varepsilon$ is very close to zero.

–    Liebovitch map: According to the proposed chaotic map,

$$x_{t+1} = \begin{cases} \alpha x_t, & 0 < x_t \leq p_1, \\ \frac{p_2 - x_t}{p_2 - p_1}, & p_1 < x_t \leq p_2 \\ 1 - \beta(1 - x_t), & p2 < x_t \leq 1 \end{cases} \tag{A11}$$

where $\alpha = \frac{p_2(1 - (p_2 - p_1))}{p_1}$ and $\beta = \frac{((p_2 - 1) - p_1(p_2 - p_1))}{p_2 - 1}$.

–    Iterative map: The definition of the iterative chaotic map with infinite collapses is as follows:

$$x_{t+1} = \sin\left(\frac{a\pi}{x_t}\right) \tag{A12}$$

where $a \in (0, 1)$.

**Appendix B. Test Functions [31]**

**Table A1.** Unimodal test functions.

| Function | Dim | Range | Shift Position | $f_{min}$ |
|---|---|---|---|---|
| $f_1(x) = \sum_{i=1}^{n} x_i^2$ | 20 | $[-100, 100]$ | $[-30, -30, \ldots, -30]$ | 0 |
| $f_2(x) = \sum_{i=1}^{n} \lvert x_i \rvert + \prod_{i=1}^{n} \lvert x_i \rvert$ | 20 | $[-10, 10]$ | $[-3, -3, \ldots, -3]$ | 0 |
| $f_3(x) = \sum_{i=1}^{n} \left(\sum_{j-1}^{i} x_j\right)^2$ | 20 | $[-100, 100]$ | $[-30, -30, \ldots, -30]$ | 0 |

**Table A1.** *Cont.*

| Function | Dim | Range | Shift Position | $f_{\min}$ |
|---|---|---|---|---|
| $f_4(x) = \max_i \{|x_i|, 1 \le i \le n\}$ | 20 | $[-100, 100]$ | $[-30, -30, \ldots, -30]$ | 0 |
| $f_5(x) = \sum_{i=1}^{n-1} [100(x_{i+1} - x_i^2)^2 + (x_i - 1)^2]$ | 20 | $[-30, 30]$ | $[-15, -15, \ldots, -15]$ | 0 |
| $f_6(x) = \sum_{i=1}^{n} ([x_i + 0.5])^2$ | 20 | $[-100, 100]$ | $[-750, \ldots, -750]$ | 0 |
| $f_7(x) = \sum_{i=1}^{n} i x_i^4 + random[0, 1)$ | 20 | $[-1.28, 1.28]$ | $[-0.25, \ldots, -0.25]$ | 0 |

**Table A2.** Multimodal test functions.

| Function | Dim | Range | Shift Position | $f_{\min}$ |
|---|---|---|---|---|
| $f_8(x) = \sum_{i=1}^{n} -x_i \sin\left(\sqrt{|x_i|}\right)$ | 20 | $[-500, 500]$ | $[-300, \ldots, -300]$ | $-418.9829 \times 5$ |
| $f_9(x) = \sum_{i=1}^{n} [x_i^2 - 10\cos(2\pi x_i) + 10]$ | 20 | $[-5.12, 5.12]$ | $[-2, -2, \ldots, -2]$ | 0 |
| $f_{10}(x) = -20\exp\left(-0.2\sqrt{\frac{1}{n}\sum_{i=1}^{n} x_i^2}\right) - \exp(\frac{1}{n}\sum_{i=1}^{n} \cos(2\pi x_i) + 20 + e)$ | 20 | $[-32, 32]$ | | 0 |
| $f_{11}(x) = \frac{1}{4000}\sum_{i=1}^{n} x_i^2 - \prod_{i=1}^{n} \cos\left(\frac{x_i}{\sqrt{i}}\right) + 1$ | 20 | $[-600, 600]$ | $[-400, \ldots, -400]$ | 0 |
| $f_{12}(x) = \frac{\pi}{n}\{10\sin(\pi y_1) + \sum_{i=1}^{n-1} (y_i - 1)^2 [1 + 10\sin^2(\pi y_{i+1})] + (y_n - 1)^2\}$ $+ \sum_{i=1}^{n} u(x_i, 10, 100, 4)$ $y_i = 1 + \frac{x_i+1}{4}$ $u = (x_i, a, k, m) = \begin{cases} k(x_i - a)^m & x_i > a \\ 0 & -a < x_i < a \\ k(-x_i - a)^m & x_i < -a \end{cases}$ | 20 | $[-50, 50]$ | $[-30, -30, \ldots, -30]$ | 0 |
| $f_{13}(x) = 0.1\{\sin^2(3\pi x_1) + \sum_{i=1}^{n} (x_i - 1)^2 [1 + \sin^2(3\pi x_i + 1)]$ $+ (x_n - 1)^2 [1 + \sin^2(2\pi x_n)]\} + \sum_{i=1}^{n} u(x_i, 5, 100, 4)$ | 20 | $[-50, 50]$ | $[-100, \ldots, -100]$ | 0 |

**Table A3.** Composite test functions.

| Function | Dim | Range | $f_{\min}$ |
|---|---|---|---|
| $F_{14}(CF1):$ $f_1, f_2, f_3, \ldots, f_{10} =$ Sphere Function $[\sigma_1, \sigma_2, \sigma_3, \ldots, \sigma_{10}] = [1, 1, 1, \ldots, 1]$ $[\lambda_1, \lambda_2, \lambda_3, \ldots, \lambda_{10}] = [5/100, 5/100, 5/100, \ldots, 5/100]$ | 10 | $[-5, 5]$ | 0 |
| $F_{15}(CF2):$ $f_1, f_2, f_3, \ldots, f_{10} =$ Griewank's Function $[\sigma_1, \sigma_2, \sigma_3, \ldots, \sigma_{10}] = [1, 1, 1, \ldots, 1]$ $[\lambda_1, \lambda_2, \lambda_3, \ldots, \lambda_{10}] = [5/100, 5/100, 5/100, \ldots, 5/100]$ | 10 | $[-5, 5]$ | 0 |
| $F_{16}(CF3):$ $f_1, f_2, f_3, \ldots, f_{10} =$ Griewank's Function $[\sigma_1, \sigma_2, \sigma_3, \ldots, \sigma_{10}] = [1, 1, 1, \ldots, 1]$ $[\lambda_1, \lambda_2, \lambda_3, \ldots, \lambda_{10}] = [1, 1, 1, \ldots, 1]$ | 10 | $[-5, 5]$ | 0 |
| $F_{17}(CF4):$ $f_1, f_2 =$ Ackley's Function $f_3, f_4 =$ Rastrigin's Function $f_5, f_6 =$ Weierstrass Function $f_7, f_8 =$ Griewank's Function $f_9, f_{10} =$ Sphere Function $[\sigma_1, \sigma_2, \sigma_3, \ldots, \sigma_{10}] = [1, 1, 1, \ldots, 1]$ $[\lambda_1, \lambda_2, \lambda_3, \ldots, \lambda_{10}] = [5/32, 5/32, 1, 1, 5/0.5, 5/0.5, 5/100, 5/100, 5/100, 5/100]$ | 10 | $[-5, 5]$ | 0 |

**Table A3.** *Cont.*

| Function | Dim | Range | $f_{\min}$ |
|---|---|---|---|
| $F_{18}(CF5)$ :<br>$f_1, f_2 =$ Rastrigin's Function<br>$f_3, f_4 =$ Weierstrass Function<br>$f_5, f_6 =$ Griewank's Function<br>$f_7, f_8 =$ Ackley's Function<br>$f_9, f_{10} =$ Sphere Function<br>$[\sigma_1, \sigma_2, \sigma_3, \ldots, \sigma_{10}] = [1, 1, 1, \ldots, 1]$<br>$[\lambda_1, \lambda_2, \lambda_3, \ldots, \lambda_{10}] = [1/5, 1/5, 5/0.5, 5/0.5, 5/100, 5/100, 5/32, 5/32, 5/100, 5/100]$ | 10 | $[-5, 5]$ | 0 |
| $F_{19}(CF6)$ :<br>$f_1, f_2 =$ Rastrigin's Function<br>$f_3, f_4 =$ Weierstrass Function<br>$f_5, f_6 =$ Griewank's Function<br>$f_7, f_8 =$ Ackley's Function<br>$f_9, f_{10} =$ Sphere Function<br>$[\sigma_1, \sigma_2, \sigma_3, \ldots, \sigma_{10}] = [0.1, 0.2, 0.3, 0.4, 0.5, 0.6, 0.7, 0.8, 0.9, 1]$<br>$[\lambda_1, \lambda_2, \lambda_3, \ldots, \lambda_{10}] = [0.1 * 1/5, 0.2 * 1/5, 0.3 * 5/0.5, 0.4 * 5/0.5, 0.5 * 5/100,$<br>$\quad 0.6 * 5/100, 0.7 * 5/32, 0.8 * 5/32, 0.9 * 5/100, 1 * 5/100]$ | 10 | $[-5, 5]$ | 0 |

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
