# Peer review of "Studying the Effect of Introducing Chaotic Search on Improving the Performance of the Sine Cosine Algorithm to Solve Optimization Problems and Nonlinear System of Equations"

_mathematics, doi:10.3390/math11051231_

Round 1
Reviewer 1 Report
This paper proposes an optimization technique to solve the NSEs,called chaotic search sine cosine algorithm (CSSCA). This study is interesting, but the current version does not meet the requirements of the journal. The following matters need to be noted.
1. the literature review needs to be rewritten and confusingly structured.
2. The contribution part of this paper does not reflect innovation.
3. Traditional SCA does not need to be presented.
4. 21 test functions need to give the source, specific form, etc.
5. The experimental part needs to be compared with SOTA algorithms.
Author Response
Reviewer(s)' Comments to Author:
Reviewer: 1
This paper proposes an optimization technique to solve the NSEs, called chaotic search sine cosine algorithm (CSSCA). This study is interesting, but the current version does not meet the requirements of the journal. The following matters need to be noted.
Reviewer#1, Concern # 1: The literature review needs to be rewritten and confusingly structured.
Author response: Thanks - We agree with the reviewer, where the literature review was rewritten.
Highlighted in the paper.
Reviewer#1, Concern # 2: The contribution part of this paper does not reflect innovation.
Author response: We agree with the reviewer, where the contribution part was modified to reflect innovation.
Highlighted in the paper.
Reviewer#1, Concern # 3: Traditional SCA does not need to be presented.
Author response: Thank for your comment. This section has been added to make the paper more complete for readers. In addition, the section of the proposed methodology is highly dependent on it
Reviewer#1, Concern # 4: 21 test functions need to give the source, specific form, etc.
Author response: Thanks - We agree with the reviewer. After revision we found that the original SCA' paper solve 19 test functions but in the available code of SCA there is 21 functions. So, the mathematical formulations of the 19 test functions are putted in the Appendix B. This is modified in the whole paper especially in results section.
Highlighted in the paper.
Reviewer#1, Concern # 5: The experimental part needs to be compared with SOTA algorithms.
Author response: Thanks - We agree with the reviewer; where another SCA-based algorithm, HGWOSCA, was included to make the comparisons unbiased. In addition, other statistical measures (non-parametric Friedman test, and Wilcoxon signed-ranks test) were used to distinguish between the three algorithms (SCA, HGWOSCA, and CSSCA).
Highlighted in the paper.
Finally, Dear reviewer:
We have addressed all the issues raised and have modified the paper accordingly. Efforts are taken to rectify the paper and significant changes have been performed to the revised manuscript and all sections were rearranged, as per your instructions and other reviewers.
We are very thankful and grateful to you for your constructive opinion on the paper and your suggestions.

Author Response
Reviewer(s)' Comments to Author:
Reviewer: 2
The review of the manuscript: mathematics- 2114906
Full Title: Studying the Effect of Introducing Chaotic Search on Improving the Performance
of the Sine Cosine Algorithm to Solve Optimization Problems and Nonlinear System of Equations
General remarks
The topic of the manuscript is attractive and might be of interest for the readers of
Mathematics. Authors suggested an optimization technique to solve 14 the NSEs, it was called
chaotic search sine cosine algorithm (CSSCA). Numerous well-known functions and many
NSEs were used to test 21 CSSCA. To demonstrate the significance of the changes made in
CSSCA, the results of 22 the recommended algorithm were contrasted with those of the original
SCA. Adding chaotic 25 search to the SCA improved results by modifying the chaotic search's
parameters.
Remarks by sections
- The Introduction section is well presented and organized. There are cited all necessary previously published papers.
- The NSE section contains one definition and considered system.
Reviewer#2, Concern # 1: Line 116, might be better. The same in Lines 117 and 121.
Author response: Thanks- We agree with the reviewer. As suggested, this point has been addressed.
Highlighted in the paper.
Reviewer#2, Concern # 2: Line 117, is enough, there is no need “for every”. Line 128, the same.
Author response: Thanks- We agree with the reviewer. As suggested, this point has been addressed.
Highlighted in the paper.
- The Proposed Methodology section contains the overview of SCA and chaos theory
and it is good written.
Reviewer#2, Concern # 3: Change × by ·
Author response: Thanks- We agree with the reviewer. As suggested, this point has been addressed in all equations.
Highlighted in the paper.
Reviewer#2, Concern # 4: Line 195, correction is necessary.
Author response: Thanks- We agree with the reviewer. As suggested, this line is rewritten and corrected.
Highlighted in the paper.
Reviewer#2, Concern # 5: Line 200, mod (1) should be erased.
Author response: Thanks- We agree with the reviewer. As suggested, this line is modified.
Highlighted in the paper.
- The Numerical Results section is clear, with good, explained Tables and Figures.
- The Conclusion section is concise.
Recommendation: manuscript is acceptable for publication after these minor changes.
Finally, Dear reviewer:
We have addressed all the issues raised and have modified the paper accordingly. Efforts are taken to rectify the paper and significant changes have been performed to the revised manuscript and all sections were rearranged, as per your instructions and other reviewers.
We are very thankful and grateful to you for your constructive opinion on the paper and your suggestions.

Reviewer 3 Report
In this paper the authors present an investigation of the combination of the sine cosine algorithm (SCA) with the chaoti search (CS), for solving non-linear systems. The combination generated the chaotic search sine cosine algorithm (CSSCA). Numerical results are presented, comparing the algorithm (CSSCA) and the traditional algorithm (SCA). The paper presents interesting and relevant results that justify its publication after corrections. - The authors must include a computational cost analysis for the two methods, expanding the discussions of comparisons between the two methods for all cases presented.
Author Response
Reviewer(s)' Comments to Author:
Reviewer: 3
Comments and Suggestions for Authors
In this paper the authors present an investigation of the combination of the sine cosine algorithm (SCA) with the chaotic search (CS), for solving non-linear systems. The combination generated the chaotic search sine cosine algorithm (CSSCA). Numerical results are presented, comparing the algorithm (CSSCA) and the traditional algorithm (SCA). The paper presents interesting and relevant results that justify its publication after corrections. The authors must include a computational cost analysis for the two methods, expanding the discussions of comparisons between the two methods for all cases presented.
Author response: Dear reviewer, Thank you for your general and important comment.
The main objective of the study is to find out the effect of introducing chaotic search (CS) on the sine cosine algorithm (SCA). When we solved the test functions, it was confirmed that the CS has an effect on improving the results by 12.71% on average, according to PD% equation. In the other part of the results, we solved the NSEs with the proposed method CSSCA and studied the effect of the CS parameters on the results. Also, after revision, we found that the original SCA' paper solve 19 test functions but in the available code of SCA there is 21 functions. So, the mathematical formulations of the 19 test functions are putted in the Appendix B. This is modified in the whole paper especially in results section. In addition, another SCA-based algorithm, HGWOSCA, was included to make the comparisons unbiased. Furthermore, other statistical measures (non-parametric Friedman test, and Wilcoxon signed-ranks test) were used to distinguish between the three algorithms (SCA, HGWOSCA, and CSSCA).
Highlighted in the paper.
Finally, Dear reviewer:
We have addressed all the issues raised and have modified the paper accordingly. Efforts are taken to rectify the paper and significant changes have been performed to the revised manuscript and all sections were rearranged, as per your instructions and other reviewers.
We are very thankful and grateful to you for your constructive opinion on the paper and your suggestions.

Reviewer 4 Report
The work entitled “Studying the Effect of Introducing Chaotic Search on Improving the Performance of the Sine Cosine Algorithm to Solve Optimization Problems and Nonlinear System of Equations” presents a numerical study for the performance of a novel NSE solving algorithm based on a two-step implementation of Chaotic Search (CS) and sine cosine algorithm (SCA) that shows an improvement of 12.45% with respect to the SCA when applied to the same set of 21 functions proposed by Mirjalili, and seven proposed functions. This work certainly holds a decent degree of academic value and presents a novel technique for solving NSEs that shows great potential for improving SCA-based algorithms. However, the presentation and the dubious arrangement of the results greatly hinder the visibility and understandability of the contributions of this work. The lack of comparison with other SCA-based algorithms, and the inferences made only with a small number of evidence to support them greatly affect their credibility. Overall, this manuscript could become a decent contribution to Mathematics only if the following major comments could be addressed.

Author Response
Reviewer(s)' Comments to Author:
Reviewer: 4
The work entitled “Studying the Effect of Introducing Chaotic Search on Improving the Performance of the Sine Cosine Algorithm to Solve Optimization Problems and Nonlinear System of Equations” presents a numerical study for the performance of a novel NSE solving algorithm based on a two-step implementation of Chaotic Search (CS) and sine cosine algorithm (SCA) that shows an improvement of 12.45% with respect to the SCA when applied to the same set of 21 functions proposed by Mirjalili, and seven proposed functions. This work certainly holds a decent degree of academic value and presents a novel technique for solving NSEs that shows great potential for improving SCA-based algorithms. However, the presentation and the dubious arrangement of the results greatly hinder the visibility and understandability of the contributions of this work. The lack of comparison with other SCA- based algorithms, and the inferences made only with a small number of evidence to support them greatly affect their credibility. Overall, this manuscript could become a decent contribution to Mathematics only if the following major comments could be addressed.
Abstract
Reviewer#4, Concern # 1: As a suggestion in line 13, authors should briefly elaborate more as to why NSEs are susceptible to being approached as an optimization problem.
Author response: Thanks - We agree with the reviewer; where we showed why NSEs are susceptible to being approached as an optimization problem.
Highlighted in the paper.
Reviewer#4, Concern # 2: Once the previous comment has been attended to, authors in lines 23 and 24 should mention some of the more important functions and NSEs used to test their algorithm.
Author response: Thanks - We agree with the reviewer; where we mentioned the functions and NSEs used to test the proposed algorithm.
Highlighted in the paper.
Reviewer#4, Concern # 3: It’s a small suggestion for the authors to consider a rewording of lines 14 and 15 to better express their intention to present the proposed name for their algorithm.
Author response: Thanks - We agree with the reviewer; where we changed the wording of lines 14 and 15 to describe the suggested algorithm more clearly.
Highlighted in the paper.
Reviewer#4, Concern # 4: A better choice for the keyword “Genetic Algorithm” should be made as it is not a key concept for the whole work to the point of being able to change it for any other NSEs solving algorithm.
Author response: Thanks - We agree with the reviewer, where the keyword “Genetic Algorithm” is deleted.
Introduction
Reviewer#4, Concern # 1: It’s suggested to add in line 35 a citation to further solidify that it has been proved to be a challenging task to find efficient and sufficiently good solutions for NSEs, or on the other hand, it’s suggested to completely reword this paragraph to avoid making a possible statement that requires solid references.
Author response: Thanks - We agree with the reviewer, where this sentence was reworded.
Highlighted in the paper.
Reviewer#4, Concern # 2: In lines 39 to 40 authors should further elaborate as to why or to whom are the Secant and Newton methods preferred and should add a correspondent reference to support this claim.
Author response: Thank you - We concur with the reviewer's suggestion to rewrite this statement and add a relevant reference to back up this assertion.
Highlighted in the paper.
Reviewer#4, Concern # 3: Authors should seriously consider simplifying or only choosing the most representative algorithms for solving NSEs as it makes the whole paragraph (from lines 48 to 60) unreadable and does not represent such an important contribution to the introduction.
Author response: Thanks – This paragraph demonstrates the most popular meta-heuristic algorithms and their advantages over traditionally approaches for solving optimization problems, NSEs included.
Reviewer#4, Concern # 4: It’s suggested to give a more synthesized description of GA, PSO, ABC, and GOA, as the operation details of these aren’t considered as important as perhaps include
examples of other models based on the SCA.
Author response: Thanks – This paragraph shows, as a survey, that meta-heuristic algorithms such as GA [23], PSO [24], ABC [25], improved cuckoo search algorithm (CSA) [26], FA [27], and GOA [28] are used to resolve NSEs.
Reviewer#4, Concern # 5: It’s of great importance for the authors to clarify why section 5 is missing in lines 109 to 112 and in its place is section 6, even though no such section exists. (If this issue just corresponds to a typo authors should fix it without further issues.)
Author response: Thanks - We agree with the reviewer. This was merely a typo that was fixed.
Highlighted in the paper.
Reviewer#4, Concern # 6: It’s recommended to the authors consider a reorganization for all the sections between sections 2 and 4 as, for example, all of sections 2 and 3 could be completely merged in a unique “Methods and Materials section” given the not-so-important contributions section 2 makes as to deserve a whole section.
Author response: Thanks - We agree with the reviewer. As suggested, sections 2 and 3 are modified.
Highlighted in the paper.
Reviewer#4, Concern # 7: A reference for line 94 should be added to solidify such a claim.
Author response: Thanks - We agree with the reviewer. As suggested, A reference for this sentence is added.
Highlighted in the paper.
Reviewer#4, Concern # 8: The list of main contributions should be visually improved so as to not hinder its legibility or altogether eliminate it and rework the whole list into a paragraph.
Author response: Thanks - We agree with the reviewer. Thanks - We agree with the reviewer. As suggested, the whole list of main contributions was reworked.
Highlighted in the paper.
Section 2 Nonlinear system of equation (NSEs):
Reviewer#4, Concern # 1: It’s suggested to add in line 114 a reference to the bibliography used to define an NSE.
Author response: Thanks - We agree with the reviewer. As suggested, a reference to the bibliography used to define an NSE is added.
Highlighted in the paper.
Section 3 The proposed methodology: Subsection 3.1 Traditional SCA
Reviewer#4, Concern # 1: It’s suggested for lines 134 to 135, to first define the abbreviature NFI before using it openly without context and to add a corresponding reference for lines 134 to 135 to give a much more solid base to the claim.
Author response: Thanks - We agree with the reviewer. As suggested, these sentences were modified by define the abbreviature NFI before using it openly without context and adding a reference.
Highlighted in the paper.
Reviewer#4, Concern # 2: Another reference should be added in lines 138-140 to provide the necessary background for such an expression as “NFL theorem” and its formal proposition.
Author response: Thanks - We agree with the reviewer. As suggested, another reference was added in these lines to provide the necessary background for such an expression and its formal proposition.
Highlighted in the paper.
Reviewer#4, Concern # 3: It’s recommended to the authors, to reword the phrase “the continuous rise in real- world issue solving” in line 135 to improve its grammatical structure and legibility.
Author response: Thanks - We agree with the reviewer. As suggested, the mentioned phrase was modified to improve its grammatical structure and legibility.
Highlighted in the paper.
Reviewer#4, Concern # 4: As minor suggestion authors should consider changing “upgrading” in line 150 to “updating” as it seems to be a more correct expression for the context in which it’s
being applied.
Author response: Thanks - We agree with the reviewer. As suggested, the mentioned whole phrase was modified and “upgrading” was changed to “updated”.
Highlighted in the paper.
Reviewer#4, Concern # 5: Authors may consider improving figure 1 in line 163, as to make it a more general scheme portraying an arbitrary ?1 value for the interval.
Author response: Thanks - We agree with the reviewer. As suggested, the figure 1 was improved to make it a more general scheme.
Highlighted in the paper.
Reviewer#4, Concern # 6: It’s strongly suggested to rewrite the whole paragraph (from lines 177 to 181) based on a much broader list of references such as Strogratz, to be able to provide a more useful and formal definition for Chaos theory and chaotic maps.
Author response: Thanks - We agree with the reviewer. As suggested, the whole paragraph was rewritten to provided a more useful and formal definition for chaos theory and chaotic maps.
Highlighted in the paper.
Reviewer#4, Concern # 7: In addition to the first correction, all of the mentioned chaotic maps are suggested to be organized in a list for this section while their more precise definitions should be
available in an annex to improve the readability of this section.
Author response: Thanks - We agree with the reviewer. As suggested, chaotic maps in this section are mentioned briefly, while the Appendix A contains more detailed definitions.
Highlighted in the paper.
Reviewer#4, Concern # 8: It’s strongly suggested to present the algorithms used for the CSSCA in the algorithm template officially used by MDPI to give more clarity and legibility to this core
section of the manuscript.
Author response: Thanks - We agree with the reviewer. The algorithms will be presented and the whole paper are put in the template officially used by MDPI in the editing version.
Reviewer#4, Concern # 9: Line 231 should be rewritten to correct its grammatical time tense.
Author response: Thanks - We agree with the reviewer, where this sentence was rewritten.
Highlighted in the paper.
Reviewer#4, Concern # 10: Word “stages” in line 233 should be changed to phases, as it’s the term used to describe both parts of the CSSCA process.
Author response: Thanks - We agree with the reviewer, where these words were modified.
Highlighted in the paper.
Section 4 Numerical Results
Reviewer#4, Concern # 1: For the list of 21 test functions mentioned in line 275, authors should include these functions and their parameters in a table in an annex with the corresponding reference to Mirjalili included as it’s a fundamental mistake to assume the notation between both works will remain the same (even though that seems the be the author’s intention) and so this addition should be a crucial addition to the completeness and credibility of the methodology employed to test the proposed algorithm.
Author response: Thanks - We agree with the reviewer. After revision we found that the original SCA' paper solve 19 test functions but in the available code of SCA there is 21 functions. So, the mathematical formulations of the 19 test functions are putted in the Appendix B. This is modified in the whole paper especially in results section.
Highlighted in the paper.
Reviewer#4, Concern # 2: It’s of great importance for numerical experiments to at least have included another SCA-based algorithm to make a more direct comparison and to use the SCA results as a control for the data, as the lack of more direct comparisons may induce biases in the interpretation of the presented results as their quantity is low. This also suggests employing a better statistical measure of dispersion between CSSCA and SCA results rather than the one presented in equation (21).
Author response: Thanks - We agree with the reviewer; where another SCA-based algorithm, HGWOSCA, was included to make the comparisons unbiased. In addition, other statistical measures (non-parametric Friedman test, and Wilcoxon signed-ranks test) were used to distinguish between the three algorithms (SCA, HGWOSCA, and CSSCA).
Highlighted in the paper.
Reviewer#4, Concern # 3: As another minor suggestion, the computer equipment in lines 280 and 281 should be displayed in a table to give a cleaner representation of this type of information.
Author response: Thanks. Only the computer equipment and program used to code the proposed algorithm are described in this sentence, and it does not need to be included in tabular form.
Reviewer#4, Concern # 4: The same case goes for the list of parameters in lines 281 to 289.
Author response: Thanks. The suggested approach has a few parameters, like any meta-heuristic algorithms, so it is not required to be presented in tabular form.
Reviewer#4, Concern # 5: It’s suggested for the sentence between lines 298 and 299 be further expanded and much more deeply justified with data from the numerical experiments, or other works that may suggest this or support this important statement as for now it remains as a weak inference made on the data obtained by the numerical experiments.
Author response: Thanks. Firstly, the aim of hybrid between SCA and CS is to escape the local minimum and advance the search process. In addition, the average improvement between the proposed algorithm (hybrid between SCA and CS) and the original SCA that obtained by PD% equation is 12.71% on average. So, this mean that we can therefore conclude that CS directs SCA to remove the local minimum and improve the search results.
Highlighted in the paper.
Reviewer#4, Concern # 6: It’s suggested to rework the aesthetics of all the tables included in this work to allow them to properly convey the information they intend.
Author response: Thanks - We agree with the reviewer, where all the tables is reworked.
Highlighted in the paper.
Reviewer#4, Concern # 7: As it was pointed out in the major comments of section 4.1, all of the present tables should be reworked to have a more clean and functional design to allow them to properly convey their information. Examples of this kind of design for tables may be seen in other works of the present MDPI Mathematics.
Author response: Thanks - We agree with the reviewer, where all the tables is reworked to the same design for tables that seen in other works of the present MDPI Mathematics.
Highlighted in the paper.
Reviewer#4, Concern # 8: Bar graphics for the table parameters, are no able to properly convey the information contained in the tables or be of any kind of visual aid to see or show any correlation beet the algorithms and the variation of their parameters. These graphics should be completely reworked or eliminated as they serve no purpose in this work, hindering its image, rather than complementing it.
Author response: Thanks. Dear reviewer, the bar graphs are a graphical presentation of tables for giving an fast and easy visualization of the effect of changing chaotic search parameters on the final results.
Reviewer#4, Concern # 9: In the case these graphics were to be improved, it’s suggested to explain why the parameter k is reduced by 0.000001 of its real value. And how this change affects the
graphs if it has any real effect.
Author response: Thanks. Dear reviewer, in the bar graphs, the parameter k is not reduced by 0.000001 of its real value. This just a scaling that was used for this chaotic search parameter to show results' differences, where it is can’t draw a bar with values of 10000 and the other bar with values of 0.00001 in the same bar graph.
Reviewer#4, Concern # 10: It’s suggested to the authors add a reference for lines 332 to 334 with the objective of explaining why those parameters were chosen.
Author response: Thanks. Dear reviewer, you know that our approach is random approach as any metaheuristic algorithm, where the improvement or accuracy is not guaranteed when solving any optimization problem. So, these parameters were chosen randomly for the proposed algorithm to study the effect of changing the parameters of CS on the results.
Highlighted in the paper.
Reviewer#4, Concern # 11: It’s extremely recommended to the authors to separate the discussion section from the results section and give the discussion its proper section or merge it with the Conclusion. This has the objective of organising the proper hierarchy of topics within the whole work, thus improving its readability.
Author response: Thanks - We agree with the reviewer. As suggested, the discussion section was merged from the conclusion section.
Highlighted in the paper.
Reviewer#4, Concern # 12: The results concerning the behaviour of the chaotic parameters shown from lines 394 to 400 should be relocated within the results section along with their respective
figures.
Author response: Thanks - We agree with the reviewer, where the results concerning the behavior of the chaotic parameters relocated within the results section along with their respective
figures.
Highlighted in the paper.
Reviewer#4, Concern # 13: Authors ought to improve their overall analysis of the results and support it with the correct amount of evidence obtained from the results presented and the sufficient high-quality references needed to support all the inferences and propositions made within this section.
Author response: Thanks - We agree with the reviewer. Another SCA-based method, HGWOSCA, was added to improve the overall analysis of the data and ensure the fairness of the comparisons. Other statistical tests (including the Wilcoxon signed-ranks test and the non-parametric Friedman test) were also employed to compare the three algorithms (SCA, HGWOSCA, and CSSCA).
Highlighted in the paper.
Reviewer#4, Concern # 14: It’s extremely recommended to the authors to elaborate more on the claim made in lines 389 and 390 and give it the proper support with sufficient references or explicit numerical evidence. What evidence leads us to think or show us that CS does indeed get rid of the local minimum for the CSA?
Author response: Thanks. As in “Reviewer#4, Concern # 5: It’s suggested for the sentence between lines 298 and 299 be further expanded and much more deeply justified with data from the numerical experiments, or other works that may suggest this or support this important statement as for now it remains as a weak inference made on the data obtained by the numerical experiments”.
The aim of the SCA/CS hybrid is to improve search efficiency and escape the local minimum. The PD% equation shows that adding CS to SCA improves the initial SCA results by 12.71%. We can therefore infer that CS instructs SCA to get rid of the local minimum and enhance the search results.
Highlighted in the paper.
Reviewer#4, Concern # 15: In lines 415 to 417, authors claim to have used several benchmark tests to measure the computational performance of the CSSCA, but authors do not include any explicit result concerning these tests. Authors should add these results or completely.
Author response: Thank you for the accuracy of the note. This sentence is a typo when making a primitive copy of the paper and it has been deleted.
Conclusion
Reviewer#4, Concern # 2: In this section, for lines 424 to 433, it is recommended to broaden the descriptions for each point in this list by giving support to certain propositions and conclusions about the work, as some of them had little to no support with experimental data or references. This is very important as some of these statements are of great significance and may prove to be an angular stone of the technique developed to produce the CSSCA.
Author response: Thanks - We agree with the reviewer. As suggested, the descriptions for each point were broadened.
Highlighted in the paper.
Reviewer#4, Concern # 2: As a general suggestion for this section, authors should explain why were the performance tests not included in the results. Then be able to provide a decent insight as to what kind of NSEs the CSSCA seems to adapt the best and if these NSEs also match with those of the CSA.
Author response: Thanks - We agree with the reviewer. In the results, CSSCA was tested by 19 test functions, the improvement was measured between it and SCA by PD% equation, and statistical measures (non-parametric Friedman test, and Wilcoxon signed-ranks test) were used to distinguish between the three algorithms (SCA, HGWOSCA, and CSSCA). In addition, this kind of NSEs is matched with those of the CSA.
Highlighted in the paper.
Reviewer#4, Concern # 3: As a final suggestion for the conclusions, the authors may want to include the present state and future direction of this research at the time of writing the article.
Author response: Thanks - We agree with the reviewer. As suggested, this section and future direction were modified.
Highlighted in the paper.
Finally, Dear reviewer:
We have addressed all the issues raised and have modified the paper accordingly. Efforts are taken to rectify the paper and significant changes have been performed to the revised manuscript and all sections were rearranged, as per your instructions and other reviewers.
We are very thankful and grateful to you for your constructive opinion on the paper and your suggestions.

Round 2
Reviewer 1 Report
The quality of the paper has been greatly improved after revision. However, there is an error in the classification of Figure 1. Is there a difference between evolutionary algorithm and swarm-based algorithm?
Author Response
Reviewer(s)' Comments to Author:
Reviewer: 1
Reviewer#1, Concern: The quality of the paper has been greatly improved after revision. However, there is an error in the classification of Figure 1. Is there a difference between evolutionary algorithm and swarm-based algorithm?
Author response: Thanks. Yes, there is a difference between evolutionary algorithm and swarm-based algorithm; where evolutionary-based algorithms are Based on biological evolution, which includes reproduction, mutation, recombination, and selection. While the swarm-based techniques are inspired from flocking of birds that mimics how swarms interact with one another and with their surroundings.
Highlighted in the paper.
Dear reviewer:
We are very thankful and grateful to you for your constructive opinion on the paper and your suggestions.

Reviewer 3 Report
The authors made all the requested corrections and clarifications, providing a version in conditions to be accepted for publication.
Author Response
Reviewer(s)' Comments to Author:
Reviewer: 3
Comments and Suggestions for Authors
The authors made all the requested corrections and clarifications, providing a version in conditions to be accepted for publication.
Dear reviewer:
We are very thankful and grateful to you for your constructive opinion on the paper and your suggestions.
